

**Multi-trace element sea surface temperature coral reconstruction for the**
**southern Mozambique Channel reveals teleconnections with the tropical**
**Atlantic**
Jens Zinke[1,2,3,4], Juan-Pablo D'Olivo[5,6], Johannes C. Gey[2], Malcolm T. McCulloch[5,6], J.
Henrich Bruggemann[7], Janice M. Lough[4,5], Mireille M. M. Guillaume[8]
[1]School of Geology, Geography and Environment, University of Leicester, LE17RH, United
Kingdom
[2]Institute for Geosciences, Freie Universitaet Berlin, Berlin, 12249, Germany
[3]Molecular and Life Sciences, Curtin University, Perth, WA, Australia
[4]Australian Institute of Marine Science, Townsville, QLD 4810, Australia
[5]The ARC Centre of Excellence for Coral Reefs Studies, Australia
[6]Oceans Graduate School and UWA Oceans Institute, The University of Western Australia,
Crawley, WA6009, Australia
[7]UMR ENTROPIE Université de La Réunion-CNRS-IRD, Saint-Denis, France &
Laboratoire d'Excellence CORAIL
[8]UMR BOrEA Muséum National d'Histoire Naturelle-SU-UCN-UA-CNRS-IRD, Paris,
France & Laboratoire d'Excellence CORAIL
*Correspondence to*: Jens Zinke (jz262@leicester.ac.uk)
**Abstract**
Here we report seasonally resolved sea surface temperatures for the southern Mozambique
Channel in the SW Indian Ocean based on multi-trace element temperatures proxy records
preserved in two *Porites* sp. coral cores. Particularly, we assess the suitability of both
separate and combined Sr/Ca and Li/Mg proxies for improved multi-element SST
reconstructions. Overall geochemical records from Europa Island *Porites* sp. highlight the
potential of Sr/Ca and Li/Mg ratios as high-resolution climate archives but also show
significant differences in their response at this Indian Ocean tropical reef site. Our
reconstruction from 1970 to 2013 using the Sr/Ca-SST proxy reveals a warming trend of 0.58
± 0.1 °C in close agreement with instrumental data (0.47 ± 0.07 °C) over the last 42 years
(1970 to 2013). In contrast the Li/Mg showed unrealistically large warming trends, most
probably caused by uncertainties around different uptake mechanisms of trace elements Li
and Mg and uncertainties in their temperature calibration. However, spatial correlations
between the combined detrended Sr/Ca, and Li/Mg proxies compared to instrumental SST at
Europa revealed robust correlations with local climate variability in the Mozambique
Channel and teleconnections to regions in the Indian Ocean and southeastern Pacific where





surface wind variability appeared to dominate the underlying pattern of SST variability. The
strongest correlation was found between our Europa SST reconstruction and instrumental
SST records from the northern tropical Atlantic SST. Only a weak correlation was found with
ENSO, with recent warm anomalies in the geochemical proxies coinciding with strong El
Niño or La Niña. We identified the Pacific/North American (PNA) atmospheric pattern,
which develops in the Pacific in response to ENSO, and the tropical North Atlantic SST as
the most likely causes of the observed teleconnections with the Mozambique Channel SST at
Europa.

## 45    1 Introduction

Ocean-atmosphere dynamics in the tropics are key drivers of large-scale climate phenomena,
such as the El Niño-Southern Oscillation (ENSO) (Angell, 1990; Trenberth et al., 1998; Xie
et al., 2010; Timmermann et al., 2018). Tropical climate variability has therefore a strong
impact on regional and global climate teleconnections. These ocean-atmosphere dynamics are
temporally variable and sensitive to small perturbations in sea surface temperature (SST)
associated with natural and anthropogenic climate change. The oceans response to the
combined effects of natural variability and greenhouse driven anthropogenic warming act at
seasonal, interannual and multi-decadal scales. The complexity of the climate system at
temporal and spatial scales; therefore, call for a comprehensive assessment of SST pattern
change in historical times (Xie et al., 2010). To investigate changing tropical climate and
model potential future scenarios, the modern climatology faces the challenge of improving
climate data coverage, especially extending the limited time length of instrumental
measurements. The earliest records of SST, measured by commercial ship traffic, mainly
along trading routes, started in the mid-19[th] century. Only with the arrival of satellite
technology in the 1980's have the oceans been covered in more detail. The limited number of
observations and drop in data quality prior to the 1980's cause considerable uncertainties in
our understanding of important climate interactions (Thompson et al., 2008; Pfeiffer et al.,
2017). These limitations make the use of paleoclimate reconstructions, particularly from
remote locations, a vital tool to learn about past climate conditions.
Coral reconstructions extending back decades to several centuries provide invaluable data to
assess past tropical climate variability (Hennekam et al., 2018; Pfeiffer et al., 2017). One of
the most robust and widely used geochemical proxies to reconstruct SST in tropical areas is



the Sr/Ca ratio from massive corals like *Porites* (Corrège, 2006; Pfeiffer et al., 2009; DeLong
et al., 2012). The Sr/Ca ratios in $CaCO_3$ precipitated during skeletal formation are negatively
correlated with temperatures, i.e. as temperatures increase, less Sr is incorporated into the
aragonite lattice relative to Ca (Alibert and McCulloch, 1997; Corrège, 2006; DeLong et al.,
2007). While the Sr/Ca proxy is a remarkably useful tool for paleoclimate reconstructions,
there are a number of limitations that need to be considered in the application of this proxy
for quantitative reconstructions. Among these, there is a significant difference in the Sr/Ca
temperature dependency of biogenic and experimentally precipitated inorganic aragonite
(Smith et al., 1979). Abiogenic aragonite has a significantly stronger Sr/Ca dependence to
temperature with a slope of -0.039 to -0.044 mmol/mol per °C than the coral skeletal Sr/Ca
with slopes ranging between -0.040 to -0.084 mmol/mol per °C (Smith et al., 1979; Cohen et
al., 2002; Gaetani & Cohen, 2006; Gaetani et al., 2011; De Carlo et al., 2015). This disparity
has been considered as the influence of strong "vital effects" during the coral
biomineralization process or bio-smoothing effects (de Villiers et al., 1995; Gagan et al.,
2012). Nevertheless, several recent studies have confirmed the suitability of the Sr/Ca
paleorecorder when carefully sampled along the optimal growth axis and ideally replicated
(Pfeiffer et al., 2009, 2017; De Long et al., 2012; Zinke et al., 2016). Recently, the use of
additional SST sensitive proxies (multi-element paleothermometry) has been tested and the
Li/Mg ratios emerged as a promising tool to reconstruct SST (Hathorne et al., 2013;
Montagna et al., 2014; D'Olivo et al., 2018). The innovation of the Li/Mg temperature proxy
lies in the normalization of Li to Mg which is thought to eliminate the influence of Raleigh
fractionation processes influencing most trace element incorporations into the coral skeleton
(Cohen et al., 2002; Gaetani & Cohen, 2006; Gaetani et al., 2011). Li/Mg was shown (e.g.
Montagna et al., 2014) to be applicable to a large range of coral species inhabiting a large
temperature range. However, to date no long-term (e.g multi-decadal) SST reconstruction has
been developed based on Li/Mg ratios to test its suitability, particularly in tropical corals, in
comparison to the established Sr/Ca time series.
The aim of this study was to reconstruct sea surface temperatures based on coral Sr/Ca,
Li/Mg and their combination to validate and extend the SST information for the southern
Mozambique Channel in the southwestern Indian Ocean. We examined instrumental climate
and coral proxy data from Europa Island. The high latitude of this atoll and the lack of human
impact make this location ideal to investigate past climate variability based on cores from
massive *Porites* corals (Fig. 1). The Mozambique Channel in the southwestern Indian Ocean



is a particular sparsely sampled region, despite its importance as a major pathway of warm
surface flow of the global ocean conveyer (De Ruijter et al., 2002; Schouten et al., 2002;
Woodruff et al., 2011; Beal et al., 2011). Furthermore, Europa Is. is located just upstream of
the region that feeds the Agulhas Current, one of the most powerful western boundary
currents on the planet. As such, this region is a possible source of both local and global
climate interactions and drivers of especially longer-term changes in patterns of SST
variability, which are investigated here.

**2 Materials and Methods**
**2.1 Study area**
Europa Is. (hereafter Europa), a 28 km$^2$ atoll that is part of the five *Eparses* Islands, of the
French *Terres Australes et Antarctiques Françaises,* and lies in the central Mozambique
Channel between southern Mozambique and southern Madagascar (22° 21' S, 40° 21' E; Fig.
1). Europa, with Hall Tablemount and Bassas da India Atoll, is part of an archipelago that
was hypothesized to have been formed by the Quathlamba hotspot, which presently lies
beneath Lesotho (Hartnady, 1985). Europa is a carbonate platform attaining a maximum
elevation of 6 m with a fossil coral terrace that was dated to the last interglacial period with
an age of approximately 94 kyr (Battistini et al., 1976), therefore formed concomitantly to
other carbonate platforms in the Mozambique Channel (Battistini et al., 1976; Guillaume et
al., 2013). The 22 km coastline is surrounded by a fringing coral reef with a fore reef slope
that dips steeply into deeper water. A geomorphological relict of the former atoll drained by
the following marine regression forms a narrow shallow lagoon that occupies more than the
half of the island and opens to the sea through the northern reef flat. A mangrove formation
ranging from shrub to forest stands grows in the salt waters of the back and windward edge of
the lagoon (Lambs et al., 2016). Europa is characterized by a high-energy environment under
the influence of south to southeasterly trade winds (strongest in austral winter) and
occasionally impacted by tropical cyclones in austral summer (Barruol et al., 2016). A train
of anticyclonic ocean eddies traveling through the Mozambique Channel transports tropical
water southward eventually feeding the Agulhas Current (Beal et al., 2011).
Europa is a no-take area; the pristine state of its fringing coral reefs was attested by scarce
macroalgae, high fish biomass and high coral coverage that locally exceeded 100% in 2013
due to superimposed platy *Acropora* stands (Guillaume & Bruggemann 2011). While almost
undisturbed from local anthropogenic disturbance, remote islands are impacted by global




change. For instance in 1998 (El Niño) a severe coral bleaching event was inferred at Europa
from a high coral mortality accompanied by a recruit cohort of small-sized *Acropora* colonies
observed 4 years later (Quod & Garnier 2004). In 2011 (La Niña), a moderate bleaching
event affecting mostly *Pocillopora* corals, massive and branching *Porites* was witnessed
(Guillaume & Bruggemann 2011).

**2.2 Instrumental temperature data**
To review and evaluate the coral geochemical proxies, various types instrumental data were
assessed. High-resolution SST data (0.25° × 0.25°) covering 1981 to 2013 was obtained from
the Advanced Very High Resolution Reconstructed Optimum Interpolation Sea Surface
Temperature version 2 (AVHRR-OISSTv2; Banzon et al., 2014, 2016; Reynolds et al.,
2007). The AVHRR-OISSTv2 dataset is composed of daily satellite data and *in situ* data,
adjusted for biases (Banzon et al., 2014). The SST data was extended back to 1970 using the
Extended Reconstructed Sea Surface Temperature (ERSSTv4; Liu et al., 2015) dataset,
which is based on the "International Comprehensive Ocean-Atmosphere Data Set"
(ICOADS; Woodruff et al., 2011). The ESRSSTv4 record is composed of satellite data from
the AVHRR data extended with temperature records from ships and buoys processed at
monthly resolution and interpolated to a spatial resolution of 2.0 degrees. In addition, *in situ*
SST data was available from April 2009 to October 2010 measured by a tide-temperature-
conductivity XR-420 6.30 RBR Ltd gauge deployed by the CNRS-INSU close to the *Porites*
coral coring site studied here (Testut et al., 2016). Finally, *in situ* air temperature records
were provided by the Meteo-France station at Europa (n° 98403003, 22.32 °S, 40.33 °E,
elevation 6 m).

**2.3 Coral core sampling, analysis and age model**
Coral cores were extracted with a pneumatic drill from two living massive *Porites* colonies
on the northeastern reef slope of Europa in early May 2013 during the *ORCIE* scientific
expedition. The cores were obtained at a depth of 12 m to 13 m in sections of ~30 cm.
Morphological identification based on skeletal features observed under an optical microscope
assigned core EU3 to *P. solida* and core EU2 to *P. mayeri*. The longest core EU3 measured
136 cm (Table 1) and was sampled from 2013 back to 1970. The shorter core EU2 was
sampled from 2013 back to 2003. Cores were sliced to 7 mm thick slabs and cleaned
following established protocols (Nagtegaal et al., 2012). The slabs were then X-rayed to




reveal the annual density banding and analysed by densitometry at the Australian Institute of
Marine Science (Figs. S1 & S2). Annual rates of linear extension were calculated from 1) X-
ray based density measurements with coral XDS software and 2) the bimonthly Sr/Ca records
by measuring the distance between Sr/Ca maxima in both records.
The basis for the optimal extraction of geochemical signals from massive *Porites* is the
precise selection of the sampling path, according to the architecture of the coral skeleton. The
X-rays show the corallite fan structures with the chosen sampling path along the central
growth axis of the corallum highlighted (Figs. S1 & S2). Carbonate powder samples of
approximately 50 mg at 2 mm continuous intervals were obtained along this sampling path
with a 0.9 mm diameter dental drill. Processing of the samples for geochemical analysis is
based on methods described by Zinke et al. (2015) and D'Olivo et al. (2018). First the
powder samples were homogenized in a small agate mortar and ~10 mg were weighed into
thoroughly cleaned 5-ml tubes. Subsequently the samples were dissolved in 0.5 ml of 0.5 N
$HNO_3$ and diluted to a calcium concentration of 100 ppm by taking an aliquot of 38 ml from
the primary dissolution and adding 3 ml of 2 % $HNO_3$ and used for the analysis of Li and
Mg. A second aliquot for the analysis of Sr, Ca, and Mg was prepared at 10 ppm by using a
300 ml aliquot of the first dilution and adding 2.7 ml of 2 % $HNO_3$ spiked with trace
concentrations of scandium, praseodymium and yttrium used as internal standards. The
analyses for trace element concentrations (TE) were made on a Thermo Scientific XSERIES
2 quadrupole ICP-MS (inductively coupled plasma mass spectrometer) at the University of
Western Australia (UWA). The Sr/Ca data reported here are normalized to the JCp-1 *Porites*
sp. standard prepared by the Geological Survey of Japan (Okai et al., 2002) with Sr/Ca =
8.838 mmol/mol. Long-term reproducibility was determined using the UWA *in-house* Davies
Reef coral standard solution with Mg/Ca = ±6.2%, Sr/Ca = ±0.4%, and U/Ca = ±1.1% (2σ
RSD; n=167) (D'Olivo et al., 2018).

The first step after generating the trace element records was to assign an age model. The 2
mm sampling resolution provided 6 to 9 samples per year for any given year and provided
robust bimonthly resolved geochemical records. Based on the instrumental SST data from
AVHRR-OI SST and the *in situ* measurements August was established the coldest month for
our location. The highest Sr/Ca annual values in the raw data were tied with the annual
minimum in the SST records using the open source time series analysis toolkit "Analyseries".



Bimonthly (6 samples per year) records were generated by linear interpolation in Analyseries
to facilitate comparisons between the different datasets.
Ordinary linear least squares regression (OLS) was used to calibrate the geochemical ratios
with the SST products. In addition to absolute SST values, SST anomalies (deviations from
bimonthly SST seasonality in any given year) were calculated relative to the reference period
of 2003 to 2013, which includes the overlap between both cores. Longer time series
anomalies were calculated relative to the 1981 to 2010 period. Bimonthly temperature
residuals were calculated for absolute SST and SST anomaly reconstructions to highlight
periods where Sr/Ca-SST and Li/Mg-SST differ from instrumental SST. To evaluate how
different trace element ratios track instrumental temperatures, uncertainties in bimonthly
absolute SST and SST anomalies for individual and composite cores were calculated based
on the root mean square error (RMSE) defined as:
$RSME = \sqrt{\frac{1}{N}\Sigma\left(T_{calc,n} - T_{meas,n}\right)^2},$

where $T_{calc,n}$ is the n'th term from the coral derived temperature and $T_{meas,n}$ is the n'th
measurement in the instrumental record and N is the total number of observations.

**3 Results**
**3.1 Bimonthly Sr/Ca, Li/Mg, Mg/Ca and Li/Ca ratios**
The bimonthly time series of all trace element ratios measured in cores EU2 and EU3 are
illustrated in Figure 2. Comparisons between the different trace element ratios for the period
of overlap between 2003 and 2012 are shown in Figure 3 and Table S1.
For the period of overlap the bimonthly time series of the Sr/Ca ratios of core EU3 ranged
between 8.78 and 9.03 mmol/mol (8.77 and 9.09 mmol/mol between 1970 and 2012) and in
EU2 between 8.90 and 9.18 mmol/mol (Fig. 2a; Fig. 3a). The mean Sr/Ca ratios of the EU2
core were ~0.1 mmol/mol higher compared to EU3, while seasonal amplitudes and trend
since 2003 were similar between cores (Fig. 2a). Sr/Ca ratios for EU2 and EU3 were highly
correlated ($r^2$ = 0.85, p < 0.001, N = 54). Both cores showed a long-term decrease (warming
trend) in Sr/Ca between 2003 and 2012. The time series of EU3 Sr/Ca between 1970 and
2012 showed a non-linear decrease starting in the mid to late-1990s with the records lowest
ratios between 1998 and 2000 and 2009 to 2011.



The seasonal range in Li/Mg ratios in core EU3 varied between 1.18 and 1.52 mmol/mol
(1.18 and 1.65 mmol/mol between 1970 and 2012) while in core EU2 it ranges between 1.35
and 1.6 mmol/mol and showed an offset to higher absolute Li/Mg ratios between 2003 and
2012 (Fig. 2b, Fig. 3a). Li/Mg between EU2 and EU3 were highly correlated ($r^2$ = 0.69, p <
0.001, N = 54), although lower than Sr/Ca ratios. The time series of EU3 Li/Mg between
1970 and 2012 showed a non-linear decrease starting in the mid to late-1990s. For the period
of overlap between 2003 and 2012, both core Li/Mg ratios showed no trend.
Mg/Ca ratios in core EU3 showed larger amplitude seasonal variations than core EU2 (Fig.
2c). EU2 Mg/Ca ratios ranged between 4.24 and 4.61 mmol/mol between 2003 and 2012
while EU3 ranged between 4.40 and 5.36 mmol/mol (4.03 and 5.36 mmol/mol between 1970
and 2012; Fig. 2c, Fig. 3b). EU2 showed lower mean Mg/Ca ratios (~0.4 mmol/mol) than
EU3 between 2003 and 2012 (Fig. 3b). Mg/Ca ratios in EU2 and EU3 were well correlated,
yet significantgly lower than Sr/Ca and Li/Mg ($r^2$ = 0.34, p < 0.001, N = 54). Overall, EU3
Mg/Ca showed an increase since 1970 with a marked switch post-2005. EU2 Mg/Ca had no
trend.
Li/Ca ratios in core EU2 ranged between 6.03 and 7.20 μmol/mol while in EU3 it ranged
between 6.04 and 6.87 μmol/mol (5.97 to 6.87 μmol/mol between 1970 and 2012; Fig. 2d,
Fig. 3c). Li/Ca ratios in EU2 and EU3 were well correlated ($r^2$ = 0.33, p < 0.001, N = 54), yet
significantly lower than Sr/Ca and Li/Mg. Li/Ca was positively correlated with Sr/Ca and
Li/Mg and negatively with Mg/Ca in both cores for most of the record (Figs. 3c, e, f; Table
S1). EU2 Li/Ca largely mirrored variations in Sr/Ca  and Li/Mg, while in EU3 Li/Ca showed
lower correlations (Table S1). EU3 interannual variability in Li/Ca deviated from the patterns
observed in the Sr/Ca, Li/Mg and Mg/Ca data in 1970/71, 1976-78, 1989/90 and between
2001 and 2004 (Fig. 2d). In those years lower EU3 Li/Ca ratios were associated with lower
Mg/Ca and higher Sr/Ca and Li/Mg ratios, opposite to the expected relationships (Fig. 2d).

**3.2 Calibration of TE/Ca and SST reconstruction**
Absolute temperature reconstructions were obtained from the regression of the bimonthly
Sr/Ca and Li/Mg ratios with the AVHRR-OISSTv2 and ERSSTv4 data (Fig. 4; Table 2; for
ERSSTv4 see Fig. S3 and for Mg/Ca and Li/Ca vs. SST see Fig. S4). Both of the coral
datasets showed highly significant (p<0.001) correlation coefficients with the temperature



products over the period of overlap (2003 to 2012) with $r^2_{EU3\ Sr/Ca}$ = 0.92, $r^2_{EU2\ Sr/Ca}$ = 0.93,
$r^2_{EU3\ Li/Mg}$ = 0.78 and $r^2_{EU2\ Li/Mg}$ = 0.93 (Table 2). The Sr/Ca and Li/Mg time series of cores
EU3 and EU2 were highly consistent in the period of overlap (2003 to 2012). EU2 Sr/Ca and
Li/Mg performed equally well in the regressions, while EU3 Li/Mg slightly underperformed
Sr/Ca (Fig. 4; Table 2). Correlation coefficients of EU3 Sr/Ca and Li/Mg for the longer
periods 1981 to 2012 and 1970 to 2012 with AVHRR-OISSTv2 and ERSSTv4, respectively,
were also high (Table 2; Fig. S5). The regression slope of TE ratios with the two SST
products varied between -0.040 and -0.051 mmol/mol per °C for Sr/Ca and between -0.045
and -0.064 mmol/mol per °C for Li/Mg (Table 2). Overall, the regression slopes were
marginally lower for regressions with AVHRR-OISSTv2 compared to ERSSTv4. Linear
OLS regressions with 1.5 years *in situ* SST data between 2009 and 2010 revealed similar
regression slopes for Sr/Ca and Li/Mg but with narrower range (-0.042 to -0.047 mmol/mol
per °C for Sr/Ca and –0.045 to -0.052 mmol/mol per °C for Li/Mg) and lower correlation
coefficients ($r^2_{EU3\ Sr/Ca}$ = 0.70, $r^2_{EU2\ Sr/Ca}$ = 0.76, $r^2_{EU3\ Li/Mg}$ = 0.73 and $r^2_{EU2\ Li/Mg}$ = 0.81). All
correlations were statistically significant with $p < 0.05$.
The maximal seasonal range over the period 1970 to 2012 of the reconstructed bimonthly
Sr/Ca-SST and Li/Mg-SST varied between 22 °C and 30 °C in both cores with a mean
seasonal amplitude of 4.33±0.67 °C (Fig. 5; for ERSST4 see Fig. S5) in close agreement with
*in-situ* SST (4.82±0.05 °C for 2009 to 2010) and regional AVHRR-OISSTv2 (4.67±0.7 °C
for 1981 to 2013) and ERSSTv4 (4.52±0.44 °C for 1970 to 2012).
Residuals (calculated as the difference between coral-derived SST and AVHRR-OISSTv2 for
individual record length) are presented in Fig. 5 and RMSE's in Table 3 (for ERSSTv4 see
Fig. S5). The coral Sr/Ca and Li/Mg-SST reconstructions had the lowest residuals between
1993 and 2012 with AVHRR-OISSTv2, with slightly larger residuals prior to 1993 (core
EU3). AVHRR-OISSTv2 displayed a more limited seasonality between 1989 and 1995
(warmer winters) with on average higher mean SST than coral-derived SST (Fig. 5; Sr/Ca,
Li/Mg and their combination). Summer SST (Sr/Ca, Li/Mg and their combination) was in
general in better agreement throughout the individual records. Sr/Ca performed best as SST
proxy followed by the combined Sr/Ca and Li/Mg-SST (Fig. 5; Table 3).
Sr/Ca, Li/Mg and SST bimonthly anomalies were calculated relative to the 2003 to 2012
(core overlap) and 1981 to 2010 (coral composite) reference period. Coral-derived SST
anomalies were calculated using the literature average proxy-SST relationships of -0.0607





mmol/mol per °C for Sr/Ca (Corrège, 2006) and -0.060 mmol/mol per °C for Li/Mg (for
*Porites* growing within 25 and 30 °C; Hathorne et al., 2013; D'Olivo et al., 2018), both
within the range of regression slopes obtained for our Europa cores. Composite coral-derived
SST anomalies were then calculated as the arithmetic mean obtained from the two cores.
Residuals were calculated as the difference between coral-derived SST anomalies and
AVHRR-OISSTv2 anomalies (for ERSSTv4 see Fig. S6).
Figure 6 illustrates the anomalies (1981 to 2010) for the composite proxy SST from the two
cores (individual cores shown in Fig. S6) compared to the SST anomalies from AVHRR-
OISSTv2 (for ERSSTv4 see Fig. S7). EU2 and EU3 Sr/Ca and Li/Mg anomalies agreed well
between records ($r_{Sr/Ca}$ = 0.46, p <0 .001, N = 57; $r_{Li/Mg}$ = 0.56, p < 0.001, N = 57; Fig. S7).
The amplitudes of the EU Sr/Ca-SST composite anomalies closely tracked AVHRR-
OISSTv2 anomalies with slightly higher residuals prior to 1993 (Fig. 6a). EU Li/Mg-SST
composite anomalies displayed similar variability as AVHRR-OISSTv2, yet the agreement
was slightly lower than for Sr/Ca-SST anomalies with a shift to lower mean Li/Mg-SST
($r_{Sr/Ca}$ = 0.37, p < 0.001, N = 189; $r_{Li/Mg}$ = 0.33, p<0.001, N = 189; Fig. 6b). The detrended
bimonthly records agreed well between composite Sr/Ca-SST and Li/Mg-SST anomalies (r =
0.73, p < 0.001, N = 252) and with AVHRR-OISSTv2 anomalies ($r_{Sr/Ca}$ = 0.41, p < 0.001, N
= 189; $r_{Li/Mg}$ = 0.31, p < 0.001, N=189). As with absolute SST, Sr/Ca-SST and Li/Mg-SST
composite anomalies showed larger residuals pre-1993 and in general cooler anomalies than
in AVHRR-OISSTv2 (Fig. 6c; for ERSSTv4 see Fig. S7). The lowest residuals were found
for Sr/Ca-SST and Sr/Ca-Li/Mg combined SST anomalies (Fig. 6c, d). RMSE's between
2003 and 2012 are lowest for Sr/Ca-SST (0.49±0.35 °C) followed by Sr/Ca-Li/Mg (0.76 ±
0.39 °C) and Li/Mg-SST (1.03±0.50 °C) while RMSE's between 1970 and 2012 are slightly
higher (Table 3).
The anomalies for the EU Sr/Ca-SST, Li/Mg-SST and Sr/Ca-Li/Mg-SST composite time
series closely tracked the anomalies in the *in situ* air temperature data (Figs. 7a-c). Sr/Ca-SST
and Sr/Ca-Li/Mg-SST performed slightly better than Li/Mg-SST ($r_{Sr/Ca}$ = 0.46, p < 0.001, N =
189; $r_{Sr/Ca-Li/Mg}$ = 0.43, p < 0.001, N = 189; $r_{Li/Mg}$ = 0.37, p < 0.001, N = 189). Air
temperatures showed marginally cooler temperature anomalies between 1970 and 1978
compared to the Sr/Ca, Li/Mg or Sr/Ca-Li/Mg composite SST anomalies. AVHRR-OISSTv2
was also in close agreement with air temperature anomalies (Fig. 7d). ERSSTv4 anomalies
mirrored air temperatures with overall slightly cooler mean SST anomalies than in air
temperatures, especially between 1970 and 1990 (Fig. 7e).




### 3.3. Coral growth parameters and SST

Linear extension rates based on the distance between annual density bands and the distance
between Sr/Ca maxima in both cores displayed interannual and multi-decadal variability (Fig.
8b; for anomalies see Fig. S8). Linear extension for EU3 was within 0.2 cm between the two
methods except during 1972 to 1973 and 1983 to 1985. The exceptionally low extension rates
during these intervals obtained in core EU3 by the density method were most probably
related to uncertainties in defining the chronology due to poorly defined density contrasts.
Therefore, for those years, the clear seasonal pattern in Sr/Ca provided a better chronology
control than X-ray densitometry. Extension rates measured with the density method in EU2
and EU3 between 1968 and 2012 showed similar values, with a mean of 1.16±0.33 cm/yr for
EU2 and 1.2±0.27 cm/yr for EU3, However, the sclerochronology based on the density
measured with X-rays had large uncertainties due to poorly defined annual density cycles in
EU2 in the older part of the record. EU2 and EU3 mean extension rates measured between
Sr/Ca maxima between 2003 and 2012 were also similar with 1.23±0.14 cm/yr for EU2 and
1.28±0.13 cm/yr for EU3 (Fig. 8). Interannual extension rates in EU2 and EU3 showed no
significant correlation.
Skeletal density in core EU3 displayed lower variability (±0.1 g/cm$^3$) than core EU2 (±0.2
g/cm$^3$) with no correlation between cores and no significant trend (Fig. 8c). Interestingly,
core EU2 displayed higher density than EU3 in El Niño years 1977/78, 1982/83, 2002/03 and
2010. EU3 and EU2 density variations were in anti-phase with extension rates for most of the
record. Variability and trends in calcification in both cores were mainly explained by changes
in extension rates.
We find no correlation between SST reconstructions or instrumental SST with either
extension or calcification rates nor skeletal density in both cores (Fig. 8a). The period of
fastest extension and highest calcification in EU3 corresponded to 1947 to 1961. No
significant changes in extension or calcification rates were associated with known El Niño
events (e.g. 1941/42 and 1998) or local cold/warm events recorded by instrumental and proxy
records (e.g. 1994, 2002).

### 3.4 Regional and large-scale climate relationships





To assess regional correlations, we compared the EU Sr/Ca-SST composite with published coral proxy records from the Mozambique Channel and their corresponding instrumental data based on ERSSTv4 (Fig. 9; Fig. S9). The proxy records included *Porites* coral oxygen isotope and Sr/Ca from Mayotte (Comoros Archipelago; 13° S, 45° E; Zinke et al., 2008) in a lagoonal setting and from Ifaty Reef in a lagoon passage (Southwest Madagascar, 23° S, 43° E; Zinke et al., 2004). Bimonthly anomalies for ERSSTv4, Sr/Ca-SST and $\delta^{18}$O-SST were calculated relative to the reference period 1973 to 1993. The ERSSTv4 records from the three sites, spanning 10° of latitude in the Mozambique Channel between 13° and 23° S, documented a statistically significant (>95%) warming trend since 1970 (Fig. 9). ERSSTv4 for 2° × 2° spatial grids near Europa and Ifaty Reef shared 98% of variability, while Mayotte shared 38% of variability with the former two sites. The coral proxy-based SST anomalies also showed statistically significant (>95%) warming trends, although generally higher than in ERSSTv4, with the exception of Ifaty that showed no trend in Sr/Ca-SST anomalies (Fig. 9).

The interannual variability in the EU Sr/Ca-SST composite anomalies and in the Mayotte and Ifaty proxy-SST anomalies fluctuated by ~±1 °C with the exception of a few warm and cold spikes in Mayotte and Ifaty time series which were not recorded at Europa (Fig. 9b-e; Fig. S9). The EU Sr/Ca-SST composite anomalies agreed best with interannual variability in Ifaty Sr/Ca-SST (r = 0.42, p < 0.001, N = 21) and $\delta^{18}$O-SST (r = 0.19, p < 0.05, N = 21), although with a cold bias in Ifaty Sr/Ca between 1985 and 1995 (Fig. 9d,e; Fig. S9). The EU Sr/Ca-SST composite anomalies and Mayotte proxy-SST anomalies showed no significant correlations, although the trend estimates were within uncertainty bounds for Mayotte $\delta^{18}$O-SST (Fig. 9b,c; Fig. S9). Mayotte Sr/Ca time series showed anomalous cold spikes between 1970 and 1978, which were not recorded in regional ERSSTv4 (Fig. S9b,c).

Spatial correlations between the EU composite SST anomalies and the AVHRR-OISSTv2 mean annual averages (July to June) between 1981 and 2012 (Fig. 10) for each grid point were calculated to investigate large-scale teleconnections. Similar spatial correlations were calculated with the AVHRR-OISSTv2 data near Europa at 22° S, 40° E (1° × 1° resolution) with the rest of the grid points (Fig. 10). The correlations of the detrended data were computed using the KNMI climate explorer (https://climexp.knmi.nl/; Trouet & Oldenborgh, 2005) with a cutoff p-value < 0.05. Similar correlation patterns were observed for the EU2/EU3 Sr/Ca-Li/Mg SST composite and the local AVHRR-OISSTv2 with global grids in



AVHRR-OISSTv2 (Figs. 10a-b). Coherent positive correlation patterns emerge in the
Mozambique Channel, the northern and southeastern Indian Ocean, the southeastern tropical
Pacific and northern tropical Atlantic. The Sr/Ca and Li/Mg ratios showed the expected
negative relationship of both proxies with regional AVHRR-OISSTv2 mirroring the SST
patterns (Figs. 10c-d). Of particular interest was the strong relationship with the northern
tropical Atlantic (0° N to 20° N, 80° W to 30° W).
The regressions of detrended coral composite Sr/Ca and Sr/Ca-Li/Mg combined SST
reconstructions with northern tropical Atlantic AVHRR-OISSTv2 revealed the strongest
positive relationships for annual means between July and June (Table 4). Correlations
between detrended coral composite Li/Mg-SST reconstructions with northern tropical
Atlantic AVHRR-OISSTv2 were slightly lower. Northern tropical Atlantic AVHRR-
OISSTv2 (1981 to 1970) and ERSSTv4 (1970 to 2017) showed positive correlations with the
Atlantic Multi-decadal Oscillation (AMO) index (r = 0.7 to 0.72, p < 0.001; Table 4). Our
detrended coral composite Sr/Ca and Sr/Ca-Li/Mg combined SST reconstructions also
showed statistically significant positive correlations with the AMO index based on ERSSTv4
(Table 4). However, AVHRR-OISSTv2 and ERSSTv4 for Europa indicated low or non-
significant correlations with the AMO, respectively, despite the strong correlations with the
northern tropical Atlantic (Table 3). Furthermore, the Tropical North Atlantic (TNA; Enfield
et al., 1999) and North Tropical Atlantic indices (NTA; Penland & Matrosova, 1998)
indicated statistically significant positive correlations with the coral-based SST and
instrumental SST data at Europa (Fig. 10e; Table 4).
The regressions of detrended seasonal averages in AVHRR-OISSTv2 for Europa with the
Niño3.4 index of ENSO variability showed weak, yet statistically significant correlations in
the season from February to April (r = 0.47; p < 0.01; Table 4). The correlations between
ERSSTv4 and Niño 3.4 were weaker (r = 0.34, p < 0.05; Table 4). The detrended coral
composite Sr/Ca, Li/Mg and their combined SST reconstructions showed no significant
correlations with the Niño3.4 index. However, the Pacific/North American (PNA) pattern
(Wallace & Gutzler, 1981), which is an atmospheric response to ENSO, showed statistically
significant correlations with AVHRR-OISSTv2 (r =0.67, p < 0.001) for Europa and the coral-
derived SST anomalies between 1981 and 2012 (r = 0.42, p = 0.014; Table 4). The spatial
correlation pattern of the PNA index with global AVHRR-OISSTv2 revealed a similar
pattern as observed for the coral-based SST (Fig. 10).






## 4 Discussion

### 4.1 Reliability of Sr/Ca and Li/Mg as SST proxies

Both of the Europa coral core Sr/Ca and Li/Mg bimonthly time series (EU3 and EU2) showed highly significant correlations with the local and regional instrumental SST products (AVHRR-OISSTv2 and ERSSTv4). For the period of overlap between 2003 and 2012 both proxies performed equally well in core EU2 while core EU3 Li/Mg slightly underperformed Sr/Ca. The regression slopes with SST were within the range of published calibrations for *Porites* corals (e.g. Hathorne et al., 2013; Montagna et al., 2014; D'Olivo et al., 2018). The bimonthly Sr/Ca, Li/Mg and combined Sr/Ca-Li/Mg absolute SST reconstructions showed small deviations (mean RMSE's between 0.45 and 0.67°C; Table 3) from the instrumental temperatures with lower winter and slightly higher summer SST. For all proxies the agreement with instrumental data was highest for the period of overlap between cores (2003 and 2012). In general, AVHRR-OISSTv2 seasonal SST amplitudes and SST anomalies showed higher correlations with the coral-based SST reconstructions than with ERSSTv4. The lower temperatures in the proxy-SST compared with the satellite data of AVHRR-OISSTv2, which measures SST at the skin of the sea surface (top few millimeters), could be related to the living depth of the corals (12 to 13 m). This was particularly apparent in several cold spikes with 2S.D. (-0.72 °C) below the 1981 to 2012 mean SST in our coral-based SST anomaly reconstructions, which were also observed in the instrumental data (1971, 1972, 1976, 1978, 1980, 1984, 1986, 1994, 2001, 2008). The most extreme cold excursions during austral summer occurred in 1986, 1994, 2001 and 2008, which were also prominent in the AVHRR-OISSTv2. 1994 stands out as the coldest anomaly between 1970 and 2012 in coral-based SST (-1.34 °C), AVHRR-OISSTv2 (-1.37 °C) and ERSSTv4 (-1.16 °C). However, January/February 1989 and March-June 1997 cold spikes exceeding 1 °C observed in AVHRR-OISSTv2 were not as extreme in our coral-based SST anomalies (-0.58 °C in 1989; -0.1 °C in 1997). A possible explanation for these cold spikes is the upwelling of colder deeper water onto the north-east coast reef promoted by the steep slopes and topography of the fore-reef (gentle sloping plain to a depth of 25m). Upwelling-related cold spikes have been recorded in temperature loggers across the Mozambique Channel at 18 m depth, potentially related to periods of active Mozambique Channel eddies interacting with the steep topography (Schouten et al., 2002; Swart et al., 2010; Van den Berg et al., 2007). Differences



between the proxy records and the instrumental records at interannual scales could also reflect limitation in the instrumental records. The ERSSTv4 data used extending back to 1970 is based on very sparse observations in the ICOADS database for the southern Mozambique Channel. The resolution of the satellite data starting in 1981 should provide the best estimates for SST near Europa; however, observations near the coast can be susceptible to biases (e.g. mixing of land temperature with SST; Brevin et al. 2017; Smit et al., 2013). The comparison between AVHRR-OISSTv2 and local air temperature anomalies revealed an excellent agreement for the years covered by the weather station and serves as a quality check for the AVHRR-OISSTv2 data for our site. Nevertheless, the absolute temperature reconstruction from the coral Sr/Ca and Li/Mg ratios showed a good agreement with the different instrumental temperature datasets. Thus, both Sr/Ca and Li/Mg provide highly reliable SST proxies and in combination have the potential to improve SST reconstructions.

The statistically significant warming-trend of $0.58 \pm 0.1$ °C (p < 0.001) between 1970 and 2012 in the coral Sr/Ca-SST composite was in close agreement with instrumental SST data ($0.47 \pm 0.07$ °C in ERSSTv4; $0.40 \pm 0.18$ °C in AVHRR-OISSTv2 since 1981). The Li/Mg-SST composite trend of $1.06 \pm 0.15$ °C (p < 0.001) is however too large, and inconsistent with both the Sr/Ca and instrumental records. The differences in warming trends in Li/Mg-SST and Sr/Ca-SST probably highlight the differences in incorporation between these elements (Montagna et al., 2014; Marchitto et al., 2018), which could be exacerbated during periods of thermal stress. The Li/Mg-SST anomalies were especially low during some years which resulted in a larger RMSE $0.67 \pm 0.65$ °C (Table 3) compared to Sr/Ca-SST and AVHRR-OISSTv2, ERSSTv4, as well as local air temperatures. A potential limitation for the use of Li/Mg as SST proxy is the small number of studies to date reporting Li/Mg-SST relationships (regression slopes) in tropical *Porites* corals (Hathorne et al., 2013; Montagna et al., 2014; D'Olivo & McCulloch, 2017; Marchitto et al., 2018; D'Olivo et al., 2018). For example, applying the mean slope of -0.049 mmol/mol per °C for marine calcifiers reported in Montagna et al. (2014) to our Li/Mg data would lead to significant overestimations of SST anomalies, hence even larger cold biases. The mean Sr/Ca-SST relationship of -0.060 mmol/mol per °C is, on the other hand, far better constrained by a much larger number of studies (e.g. Corrège 2006; De Long et al., 2012; Pfeiffer et al., 2017). In particular, EU3 Li/Mg was most likely affected by uncertainties in the incorporation of Mg and Li into the skeleton while EU2 Li/Mg showed no irregularities. Interestingly on a seasonal scale, Li/Ca and Mg/Ca in EU3 showed the expected negative correlation; however, on interannual to



decadal scales these ratios were positively correlated. This perhaps reflects an "extreme" example of growth effects on Li and Mg unrelated to temperature (e.g. the effect of cation entrapment and heterogeneous distribution in the centres of calcification; Montagna et al., 2014; Marchitto et al., 2018). In most cases calculating Li/Mg cancels out this effect by leaving SST as the main controlling parameter; however, in core EU3 this appears not to be as effective, leading to slightly lower correlation of Li/Mg with SST compared to Sr/Ca with SST. Careful inspection of the sampling path along the major growth axes revealed that a potential cause for discrepancies in Li/Ca and Mg/Ca affecting Li/Mg might be due to suboptimal sampling along parallel growth axes. The overlap period of 2003 to 2012 was sampled continuously in both cores without switch in growth axis, showing an excellent agreement between cores including Li/Ca (r=0.57, p<0.001, N=54) and Mg/Ca (r=0.58, p<0.001, N=54). Prior to 2003, core EU3 was sampled along three different growth axes (1970 to 1982, 1983 to 1996 and 1996 to 2002) orientated at an angle along the core length. Although all axes showed optimal growth orientations, the Li/Ca ratios in EU3 deviated from the trends shown in Mg/Ca, Sr/Ca and Li/Mg. The importance of optimal sampling along continuous main growth axes for optimal TE ratios and stable isotope determinations has been shown in several recent studies (e.g. De Long et al., 2012; Zinke et al., 2016), which also appears to be the case for the Li/Mg proxy. This sensitivity of the Li and Mg proxies in core EU3, identified as *P. solida*, could also reflect a species due to slight differences in their calcification strategies. For example, D'Olivo *et al*., (2018) showed a deviation for *Porites solida* corals from other massive *Porites* species in the relationships between Sr/Ca and Li/Mg with temperature. However, this requires further investigations as this is the first long-term (multi-decadal) reconstruction based on Porites corals. Despite these uncertainties Li/Mg was overall the second-best performing proxy in this study with the detrended Li/Mg data showing an excellent agreement with the instrumental SST and Sr/Ca data. Furthermore, the interannual and decadal SST variations as well as spatial correlation patterns in the Li/Mg appeared not to have been affected and can be interpreted with high confidence as indicated by our field correlations. Overall the results from this study confirms that Sr/Ca and Li/Mg SST proxies are the most reliable proxies to date and in combination can provide with greater confidence, more reliable SST reconstructions (D'Olivo et al., 2018).

## 4.2 Regional and large-scale climate teleconnections

The spatial correlations between Europa composite data and global SST data indicated a strong response to local variability in the Mozambique Channel at the latitude between 15 °S



and 30 °S. The pattern of spatial correlation also suggested teleconnections with the
northern/eastern Indian Ocean, the southeastern Pacific and the tropical Atlantic. ENSO
influence in the instrumental data was weak and absent in our proxy records. Only the
warmest years (summer maxima) of the Europa composite time series corresponded with
strong El Niño (1998, 2010) and La Niña (1999, 2000, 2011) events, as attested for example
by the high coral mortality reported from 1998 (Quod & Garnier 2004) and the moderate
coral bleaching observed in 2011 (Guillaume & Bruggemann 2011). Local air temperatures
and AVHRR-OISSTv2 anomalies indicated other warm years (>0.5 °C) corresponding to El
Niño years (1983, 1988, 1991/92), La Niña years (1989, 1996) and ENSO neutral years
(1981, 2007, 2012/13). The majority of these lower magnitude warm events were also
recorded in the coral proxy time series. Overall these results suggest a weak or variable
impact of ENSO around Europa.
Perhaps the most interesting and to some extent unexpected relationship of our study region
was found with the northern tropical Atlantic (5 °N to 20 °N, 30 °W to 80 °W), a region that
corresponds with the main development region for Hurricanes in the tropical Atlantic
(Knutson et al., 2010). This region also has strong relationships with the AMO, which is the
leading mode of multi-decadal variability in the northern Atlantic and thought to be driven by
Atlantic Meridional Ocean Circulation (AMOC) variability (e.g. Schlesinger & Ramankutty,
1994; Kerr, 2000; Knight et al., 2005). Our coral-based SST reconstructions and satellite data
revealed a strong relationship with both the Tropical North Atlantic (TNA; Enfield et al.,
1999) and North Tropical Atlantic indices (NTA; Penland & Matrosova, 1998) as well as the
AMO since at least 1970 (Table 4). However, the ERSSTv4 for Europa showed non-
significant correlations with the AMO, while the relationship with the northern tropical
Atlantic SST was robust. The exact mechanism for this teleconnection between the
Mozambique Channel and the tropical Atlantic remains elusive. We speculate that
atmospheric processes in response to AMO, tropical Atlantic or Indo-Pacific variability
might be controlling this relationship since all correlated regions lie within or near trade wind
convergence zones (the Intertropical Convergence Zone) where atmospheric circulation
associated with deep convection controls underlying SST (Schott et al., 2009; Xie et al.,
2010; Marshall et al., 2014; Green et al., 2017; Koseki & Bhatt, 2018). Wind-driven
upwelling controls the regions with positive correlations (negative with the geochemical
proxies) in the northern and eastern Indian Ocean, the northern South China Sea and the
southeastern Pacific (Schott et al., 2009; Xie et al., 2009; Varela et al., 2015; Sydeman et al.,





2014). The strongest resemblance to an atmospheric pattern driving the observed SST pattern, including teleconnections with the Mozambique Channel, was found with the Pacific/North American (PNA) pattern (Trenberth et al., 1998). The PNA is one of the strongest modes of low-frequency atmospheric variability in the Northern Hemisphere with an equivalent Pacific/South American pattern in the Southern Hemisphere (Mo & Peagle, 2001; Irving & Simmonds, 2016). The PNA/PSA pattern is strongly influenced by ENSO and the Pacific Decadal Oscillation (PDO; Mantua et al., 1997), tending towards being in its positive phase during El Niño and negative phase during La Niña (Rodionov & Assel, 2001). The importance of atmospheric processes for Indo-Pacific climate teleconnections emanating from the Pacific PNA and PSA patterns has been documented in several studies (Rodionov & Assel, 2001; Dai et al., 2017). The Mozambique Channel and adjacent southern Africa are both impacted by PNA/PSA variability (Blamey et al., 2018). Therefore, our spatial correlation between the TNA, NTA, PNA and global SST point towards tropical-extratropical atmospheric forcing of the observed SST teleconnection patterns in our study.

**5 Conclusions**

A comparison of multiple trace element ratios in two *Porites* cores from Europa (southern Mozambique Channel) indicated that Sr/Ca was the most robust paleothermometer analysed. In addition to Sr/Ca, Li/Mg and their combination showed great potential for improved higher confidence multi-element SST reconstructions. The SST over the last 42 years (1970 to 2012) was dominated by interannual variability with a warming trend of 0.58 ± 0.1 °C in Sr/Ca-SST in close agreement with instrumental data (0.47 ± 0.07 °C). Li/Mg and the combination of Li/Mg and Sr/Ca showed unrealistically large warming trends, most probably caused by uncertainties around Li/Ca and Mg/Ca incorporation with marginally different uptake mechanisms for these trace elements. However, detrended data from Sr/Ca, Li/Mg and the combination of Li/Mg and Sr/Ca agreed well with each other and with regional instrumental SST and local air temperature. Spatial correlations between detrended Sr/Ca, Li/Mg and combined proxies with instrumental SST at Europa revealed robust correlations with local climate variability in the Mozambique Channel and teleconnections to regions in the tropical Atlantic Ocean, Indian Ocean and southeastern Pacific where surface wind variability appeared to dominate the underlying SST. Of particular interest is the strong correlation found between the proxy and instrumental SST records with the northern tropical Atlantic SST. Only a weak correlation was found with ENSO, with recent warm anomalies in the geochemical proxies coinciding with strong El Niño or La Niña. We identified the PNA





atmospheric pattern, which develops in the Pacific in response to ENSO, and the tropical
North Atlantic SST as the most likely causes of the observed teleconnections with the
Mozambique Channel SST. In conclusion, the Europa *Porites* sp. geochemical records
highlight the great potential of Sr/Ca and Li/Mg ratios as accurate, reliable high-resolution
climate archives for the tropical oceans.
**6 Data availability**
Trace element data will be made publically available on the NOAA's WDC paleoclimate data
server https://www.ncdc.noaa.gov/data-access/paleoclimatology-data/datasets.

**7 Author contribution**
JZ, MMMG, JHB, JCG and JPD designed the study and lead the writing of the manuscript.
MMMG provided the samples, JZ, JPD and MMM organised and performed the trace
element analysis, while JML did the coral densitometry measurements. All co-authors
contributed to analysis and writing of the manuscript.

**8 Acknowledgments**
The scientific expedition *ORCIE 2013* conducted by Mireille Guillaume benefited from
financial support from CNRS-INEE for the inter-organism program *îles Eparses* and from the
*Association Française des Plongeurs Scientifiques* (COLIMPHA). Authorisations for diving
around Europa and a CITES export permit (# FR1398400001-E) were provided by the *Terres*
*australes et antarctiques françaises* (TAAF) administration. The assistance of the skipper J-B
Galves and crew of the vessel *Inventive* is gratefully acknowledged. We further thank the
professional divers Jean-Patrick Rousse and Erwan Meyer for their efficient help in coring
the coral colonies, and the Division Technique from CNRS-INSU, especially Michel Calzas,
Christine Drezen and Christophe Guillerm, for sharing the temperature records of the RBR
gauges, that were deployed and retrieved by Jean-Patrick Rousse. MMMG also received
funds from the MNHN/UMR BOrEA and from the ANR-STORISK project (No.ANR-15-
CE03-0003) for element analysis. We thank Kai Rankenburg from The University of Western
Australia Advanced Geochemical Facility for Indian Ocean Research for support in trace
element measurements. Météo-France is acknowledged. Research conducted at UWA was
supported by the Australian Research Council through the Centre of Excellence for Coral
Reef Studies (CE140100020), and a Laureate Fellowship awarded to Malcolm McCulloch
(FL120100049).



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



630 **Tables**

| Location | Latitude (S) | Longitude (E) | Depth (m) | Core # | Length | Collection dates |
|---|---|---|---|---|---|---|
| **North Reef** | 22°19.839 | 40°21.758 | 12.80 | EU-2 | 105 | 2/5/2013 |
| **North-East Reef** | 22°20.119 | 40°23.333 | 12.00 | EU-3 | 136 | 3/5/2013 |

Table 1 – Coral core GPS locations from Europa, water depth, core name, core length and collection dates.




| core | proxy | SST product | slope | Conf. interval | intercept | Conf. interval | $r^2$ | $r^2$ adj. | SSE | RMSE | DoF | Period |
|------|-------|-------------|-------|----------------|-----------|----------------|-------|------------|-----|------|-----|--------|
| **EU2** | Sr/Ca | AVHRR-OISSTv2 | -0.045 | 0.0031 | 10.241 | 0.083 | 0.093 | 0.94 | 0.021 | 0.02 | 53 | 2003-2012 |
| | | ERSSTv4 | -0.051 | 0.0038 | 10.395 | 0.100 | 0.93 | 0.93 | 0.023 | 0.021 | 53 | 2003-2012 |
| | Li/Mg | AVHRR-OISSTv2 | -0.045 | 0.0033 | 2.672 | 0.088 | 0.93 | 0.93 | 0.023 | 0.021 | 53 | 2003-2012 |
| | | ERSSTv4 | -0.051 | 0.0044 | 2.815 | 0.115 | 0.91 | 0.91 | 0.030 | 0.024 | 53 | 2003-2012 |
| **EU3** | Sr/Ca | AVHRR-OISSTv2 | -0.04 | 0.0031 | 9.974 | 0.083 | 0.92 | 0.92 | 0.020 | 0.019 | 53 | 2003-2012 |
| | | ERSSTv4 | -0.046 | 0.0035 | 10.117 | 0.093 | 0.92 | 0.92 | 0.020 | 0.019 | 53 | 2003-2012 |
| | Li/Mg | AVHRR-OISSTv2 | -0.052 | 0.007 | 2.739 | 0.200 | 0.78 | 0.78 | 0.119 | 0.047 | 53 | 2003-2012 |
| | | ERSSTv4 | -0.060 | 0.009 | 2.927 | 0.226 | 0.78 | 0.78 | 0.119 | 0.047 | 53 | 2003-2012 |
| | Sr/Ca | AVHRR-OISSTv2 | -0.042 | 0.002 | 10.049 | 0.064 | 0.86 | 0.86 | 0.138 | 0.027 | 184 | 1981-2012 |
| | | ERSSTv4 | -0.048 | 0.002 | 10.191 | 0.058 | 0.88 | 0.88 | 0.179 | 0.027 | 252 | 1970-2012 |
| | Li/Mg | AVHRR-OISSTv2 | -0.057 | 0.004 | 2.898 | 0.117 | 0.77 | 0.77 | 0.452 | 0.049 | 184 | 1981-2012 |
| | | ERSSTv4 | -0.064 | 0.004 | 3.076 | 0.108 | 0.78 | 0.78 | 0.612 | 0.049 | 252 | 1970-2012 |

**Table 2** – Linear optimal least squares regression equations for core EU2 and EU3 Sr/Ca and Li/Mg ratios with AVHRR-OI SSTv2 and ERSSTv4. Conf. interval= 95% confidence interval of the regression slopes and intercepts; $r^2$ adj.= $r^2$ adjusted; SSE= Standard Error; RMSE= Root Mean Square Error; DoF= degrees of freedom.





| Core ID | Individual proxies RMSE and SD | | | | | |
|---------|---------|---------|---------|---------|---------|---------|
| | **Sr/Ca** | **Li/Mg** | **Mg/Ca** | **Li/Ca** | **Sr/Ca-Li/Mg** | **Period** |
| EU2 | 0.38±0.26 | 0.36±0.30 | 1.13±0.89 | 0.58±0.50 | 0.32±0.25 | 2003-2012 |
| EU3 | 0.41±0.29 | 0.90±0.99 | 1.79±1.76 | 1.65±1.22 | 0.60±0.57 | 2003-2012 |
| EU3 | 0.55±0.36 | 0.74±0.66 | 1.33±1.18 | 1.41±0.98 | 0.62±0.50 | 1981-2012 |
| Avg. all | **0.45** | **0.67** | **1.42** | **1.21** | **0.51** | 1981-2012 |
| S.D. all | **0.30** | **0.65** | **1.28** | **0.90** | **0.44** | 1981-2012 |

**Table 3** – Root mean square error (RMSE) and their standard deviation (S.D.) for trace element ratios against AVHRR-OISSTv2 for individual trace element ratios and Sr/Ca-Li/Mg combination. Period used for calculation of RMSE indicated in last column.



| | Northern Tropical Atlantic SST | AMO Index | TNA/NTA | Niño3.4 |
|---|---|---|---|---|
| #EU-AVHRR-OISSTv2 | 0.59*** | 0.37** | 0.55***/ 0.53*** | 0.47** (FMA) |
| ^EU-ERSSRv4 | 0.37** | 0.18 | 0.34**/ 0.36** | 0.34** (JFMA) |
| ^EU-composite Sr/Ca SST anomalies | 0.61*** | 0.46*** | 0.53***/ 0.45** | - |
| ^EU-composite Li/Mg SST anomalies | 0.55*** | 0.54*** | 0.40** / 0.32* | - |
| ^EU-composite Sr/Ca-Li/Mg SST anomalies | 0.60*** | 0.52*** | 0.47***/ 0.39** | - |

#1981-2013; ^1970-2013; *=90%, **=95%, ***=99% significance

**Table 4** – Linear correlation of detrended, mean annual instrumental and coral proxy-based SST for Europa (EU) with northern tropical Atlantic SST, the AMO index based on ERSSTv4, the Tropical North Atlantic (TNA; Enfield et al., 1999) and North Tropical Atlantic index (Penland & Matrosova, 1998) and the seasonal Niño3.4 index (Kaplan et al., 1998).





**Figure captions**

Fig. 1 Coral collection sites for cores EU2 and EU3 along the northern-northeastern reef slope of Europa and its positioning within the southern Mozambique Channel (south-west Indian Ocean).

Fig 2 Bimonthly interpolated time series of trace element/Ca proxies from cores EU2 and EU3. a) Sr/Ca, b) Li/Mg, c) Mg/Ca and d) Li/Ca.

Figure 3 – Scatter plot of bimonthly trace element ratios in cores EU2 (black dots) and EU3 (blue dots) over the full length of the records. a-c) Sr/Ca ratios vs. Li/Mg, Mg/Ca and Li/Ca, d-e) Li/Mg vs. Mg/Ca and Li/Ca and f) Li/Ca vs. Mg/Ca. The 95% prediction intervals of the regressions are indicated by red dashed (EU3) and green solid lines (EU2) and linear fits for each core with a red line. Regression equations are provided in Table S1.

Figure 4 - Linear regressions of TE/Ca proxies with AVHRR-OISSTv2 (Banzon et al., 2016) for core EU3 1981-2012 (a,c) and EU2 2003-2012 (b,d). The TE/Ca records were calibrated using the respective linear regression equations of the bimonthly correlations obtained for each of the core records from the two sites. The 95% confidence intervals of the regressions are indicated. Regression equations are provided in Table 2.

Figure 5 Absolute SST reconstructions for cores EU3 (red) and EU2 (blue) with SST residuals based on the calibration period 1981 to 2012 for a) Sr/Ca-SST, b) Li/Mg-SST and c) their combination in comparison to AVHRR-OISSTv2 (Banzon et al., 2016; black) and *in situ* SST (orange; 2009-2010). d) residuals for Sr/Ca-SST, Li/Mg-SST and their combination for cores EU2 and EU3 with respect to the AVHRR-OISSTv2 data (Banzon et al., 2016).

Figure 6 - SST anomaly reconstructions with SST residuals for a) EU composite Sr/Ca, b) EU composite Li/Mg and c) their combination for cores EU2 and EU3. d) residuals for SST anomalies of Sr/Ca-SST, Li/Mg-SST and their combination for cores EU2 and EU3 with respect to the AVHRR-OISSTv2 data (Banzon et al., 2016). Anomalies were calculated relative to the 1981 to 2010 average bimonthly seasonal cycle.

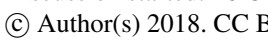


Figure 7 Comparison of coral composite Sr/Ca-SST, Li/Mg-SST and Sr/Ca-Li/Mg-SST anomaly reconstructions with air temperature from Europa Météo-France weather station data. a) Sr/Ca-SST composite, b) Li/Mg-SST composite, c) Sr/Ca-Li/Mg-SST composite, d) Europa gridded AVHRR-OISSTv2 (Banzon et al., 2016) and e) Europa gridded ERSSTv4. Anomalies were calculated relative to the 1981 to 2010 average bimonthly seasonal cycle for proxy reconstructions, instrumental SST and air temperatures.

Figure 8 - Mean annual coral growth parameters of cores EU2 and EU3 compared to coral composite Sr/Ca-SST reconstruction, AVHRR-OISSTv2 (Banzon et al., 2016) and ERSSTv4 (Liu et al., 2015). a) mean annual SST time series, b) linear extension rate, c) skeletal density and d) calcification rate.

Figure 9 - Regional comparison of Mozambique Channel ERSSTv4 anomalies for Mayotte (green), Europa (orange) and Ifaty Madagascar (blue) in a) with linear warming trends in brackets. EU Sr/Ca-SST composite anomaly compared with b) Mayotte $\delta^{18}$O-SST anomaly (blue), c) Mayotte Sr/Ca-SST anomaly (blue), d) Ifaty $\delta^{18}$O -SST anomaly (blue) and e) Ifaty Sr/Ca-SST anomaly (blue). Anomalies were calculated for the 1973 to 1993 reference period. Linear warming trends indicated in b) to e) for proxy-SST for individual record length with EU composite Sr/Ca-SST anomaly only indicated once in panel b. Proxy data taken from Zinke et al. (2004, 2008).

Figure 10 - Spatial correlations of proxy-based coral composite SST reconstructions with local and global AVHRR-OISSTv2 for mean annual data (Banzon et al., 2016). a) local AVHRR-OISSTv2 with global AVHRR-OISSTv2, b) local AVHRR-OISSTv2 with EU composite Sr/Ca-Li/Mg-SST, c) local AVHRR-OISSTv2 with EU composite Sr/Ca ratios and d) local AVHRR-OISSTv2 with EU composite Li/Mg ratios. Panels e and d show spatial correlations of the TNA and PNA indices with global AVHRR-OISSTv2. Only correlations with p<0.05 were coloured.





**Figures**

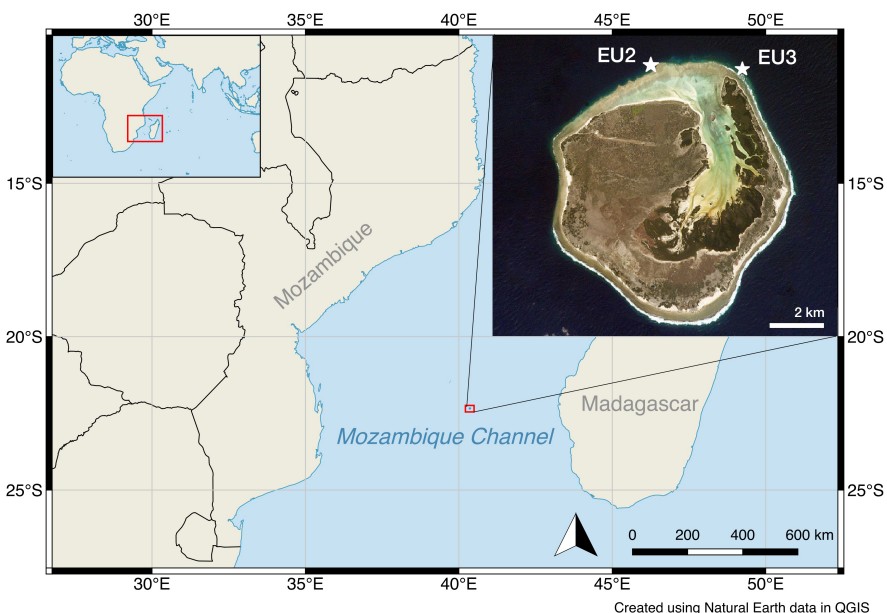

Created using Natural Earth data in QGIS

Fig. 1 Coral collection sites for cores EU2 and EU3 along the northern-northeastern reef slope of Europa and its positioning within the southern Mozambique Channel (south-west Indian Ocean).



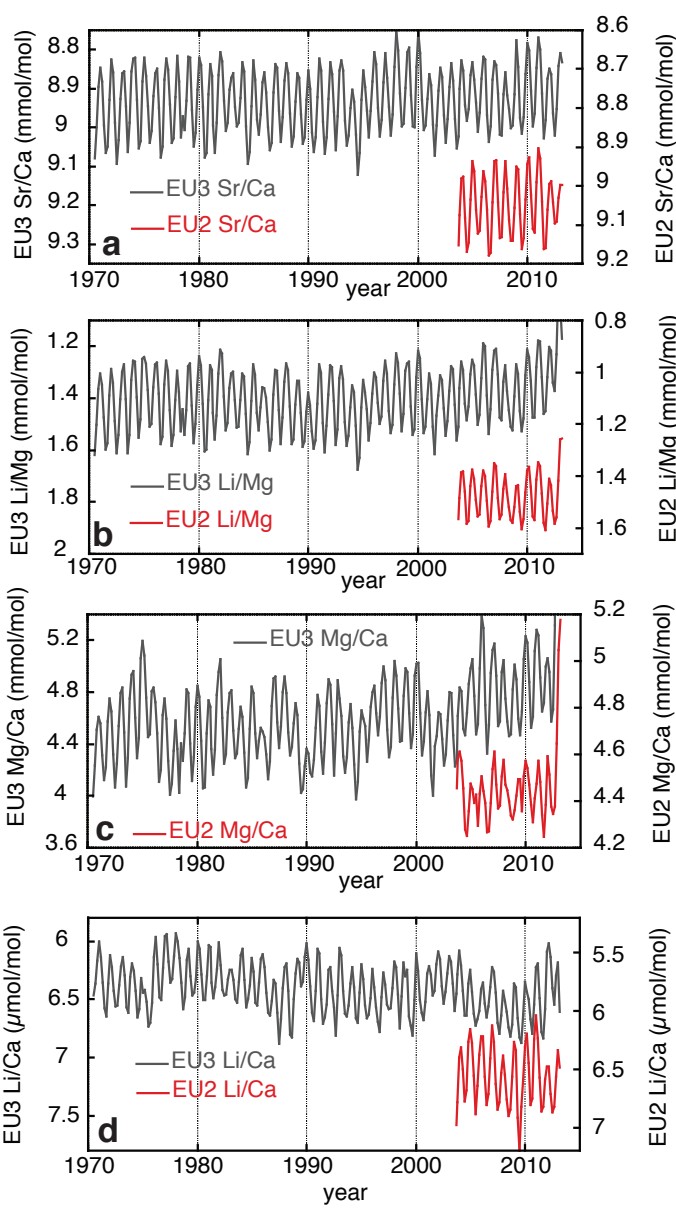

Fig 2 Bimonthly interpolated time series of trace element/Ca proxies from cores EU3 and EU2. a) Sr/Ca, b) Li/Mg, c) Mg/Ca, d) Li/Ca.





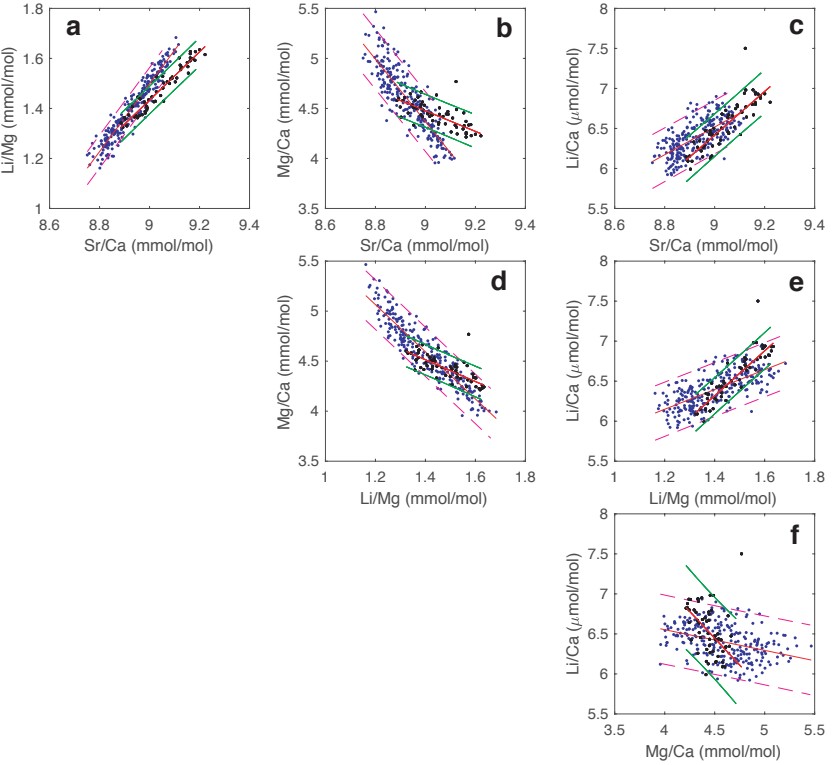

Figure 3 – Scatter plot of bimonthly trace element ratios in cores EU2 (black dots) and EU3 (blue dots) over the full length of the records. a-c) Sr/Ca ratios vs. Li/Mg, Mg/Ca and Li/Ca, d-e) Li/Mg vs. Mg/Ca and Li/Ca and f) Li/Ca vs. Mg/Ca. The 95% prediction intervals of the regressions are indicated by red dashed (EU3) and green solid lines (EU2) and linear fits for each core with a red line. Regression equations are provided in Table S1.





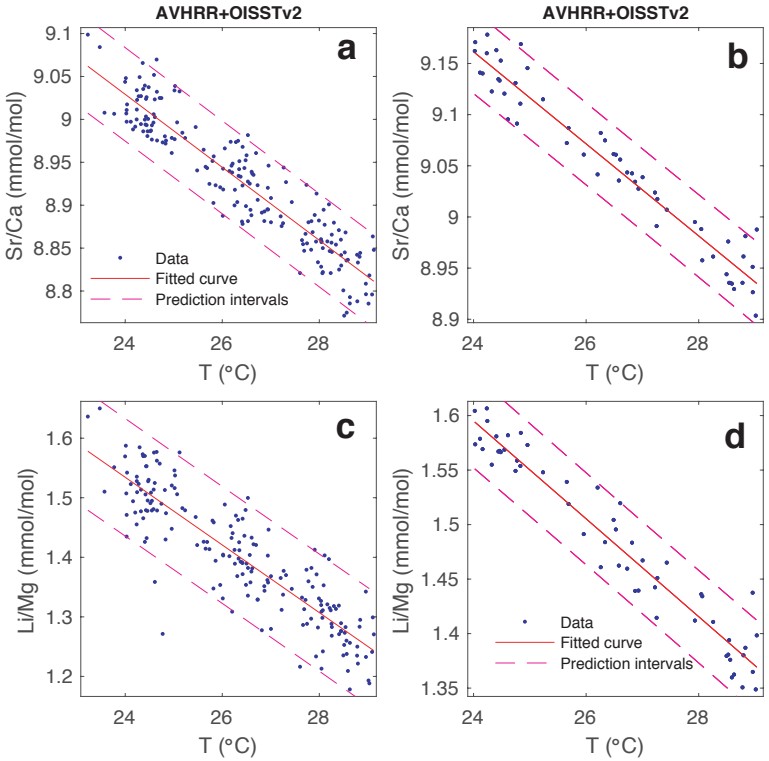

Figure 4 - Linear regressions of TE/Ca proxies with AVHRR-OISSTv2 for core EU3 1981-2012 (a,c) and EU2 2003-2012 (b,d). The TE/Ca records were calibrated using the respective linear regression equations of the bimonthly correlations obtained for each of the core records from the two sites. The 95% confidence intervals of the regressions are indicated. Regression equations are provided in Table 2.



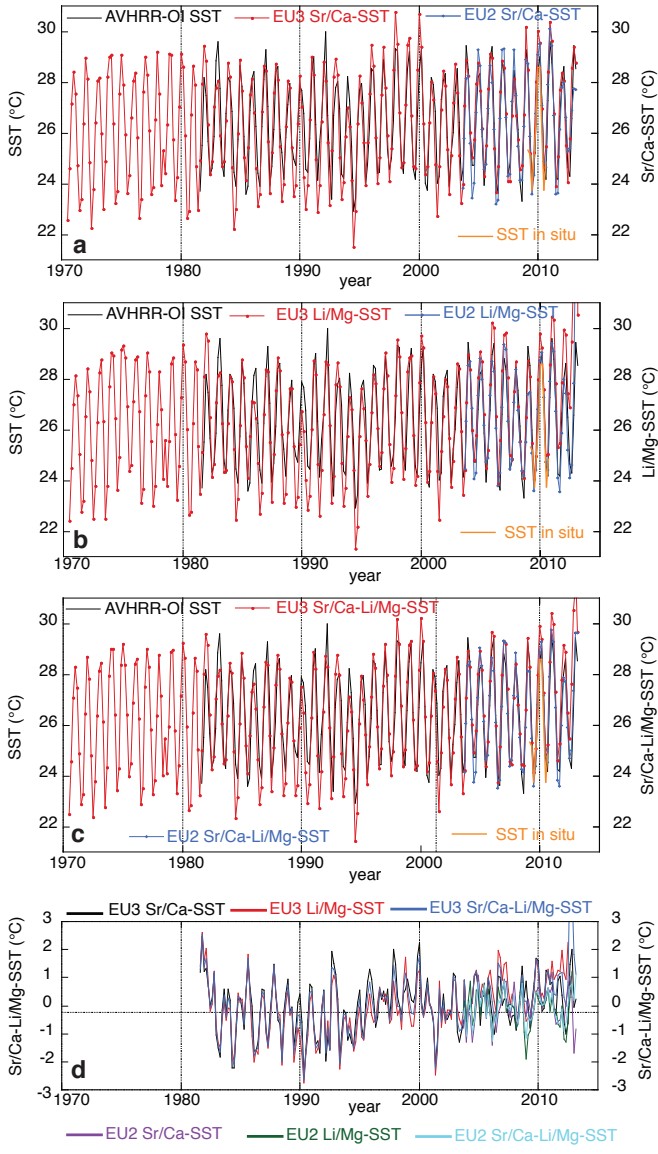

Figure 5 Absolute SST reconstructions for cores EU3 (red) and EU2 (blue) with SST residuals based on the calibration period 1981 to 2013 for a) Sr/Ca-SST, b) Li/Mg-SST and c) their combination in comparison to AVHRR-OISSTv2 (Banzon et al., 2016; black) and *in situ* SST (orange; 2009-2010). d) residuals for Sr/Ca-SST, Li/Mg-SST and their combination for cores EU2 and EU3 with respect to the AVHRR-OISSTv2 data.


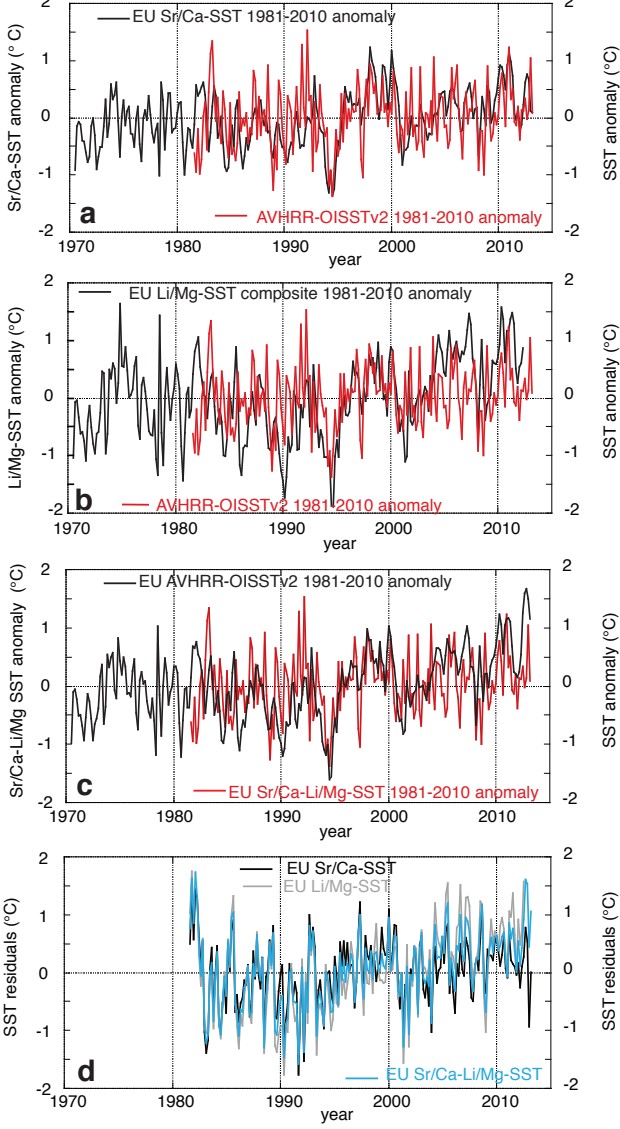

Figure 6 - SST anomaly reconstructions with SST residuals for a) EU composite Sr/Ca, b) EU composite Li/Mg and c) their combination for cores EU2 and EU3. d) residuals for SST anomalies of Sr/Ca-SST, Li/Mg-SST and their combination for cores EU2 and EU3 with respect to the AVHRR-OISSTv2 data (Banzon et al., 2016). Anomalies were calculated relative to the 1981 to 2010 average bimonthly seasonal cycle.





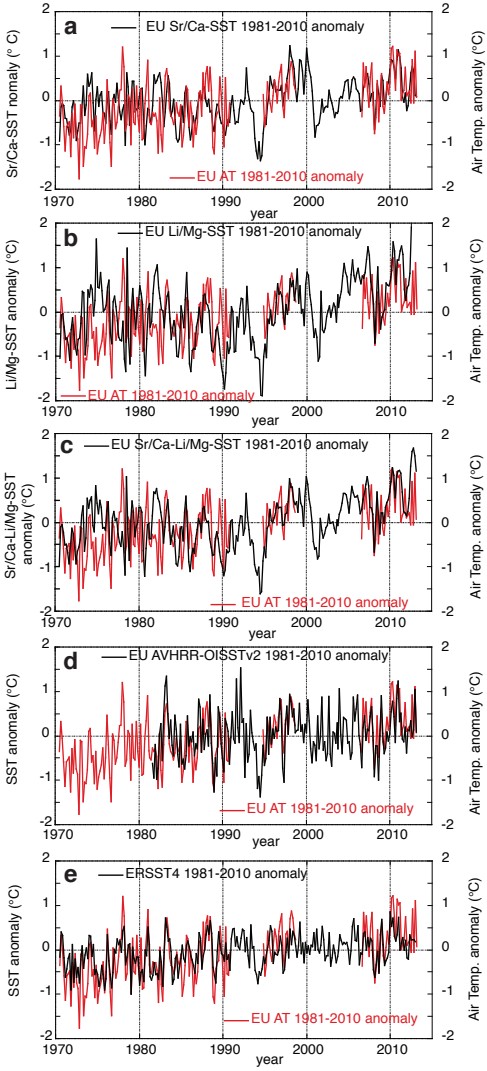

Figure 7 Comparison of Europa (EU) coral composite Sr/Ca-SST, Li/Mg-SST and Sr/Ca-Li/Mg-SST anomaly reconstructions with air temperature (AT) from Europa Météo-France weather station data. a) Sr/Ca-SST composite, b) Li/Mg-SST composite, c) Sr/Ca-Li/Mg-SST composite, d) Europa gridded AVHRR-OISSTv2 (Banzon et al., 2016) and e) Europa gridded ERSSTv4 (Liu et al., 2015). Anomalies were calculated relative to the 1981 to 2010 average bimonthly seasonal cycle for proxy reconstructions, instrumental SST and air temperatures.



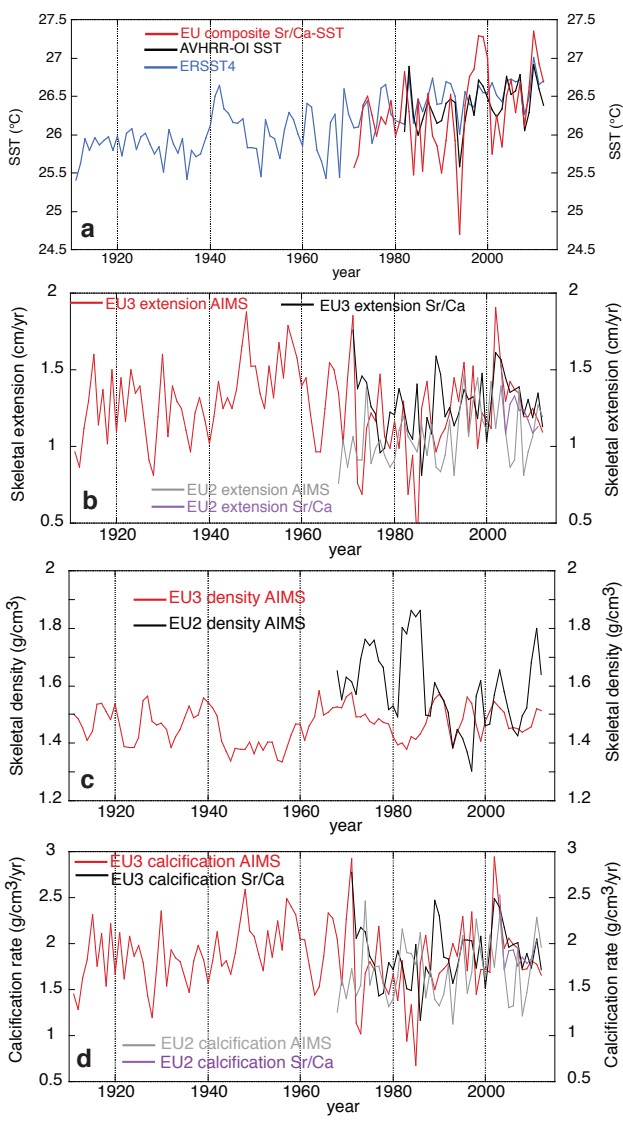

Figure 8 - Mean annual coral growth parameters of cores EU3 and EU2 compared to coral composite Sr/Ca-SST reconstruction, AVHRR-OISSTv2 (Banzon et al., 2016) and ERSSTv4 (Liu et al., 2015). a) mean annual SST time series, b) linear extension rate, c) skeletal density and d) calcification rate.





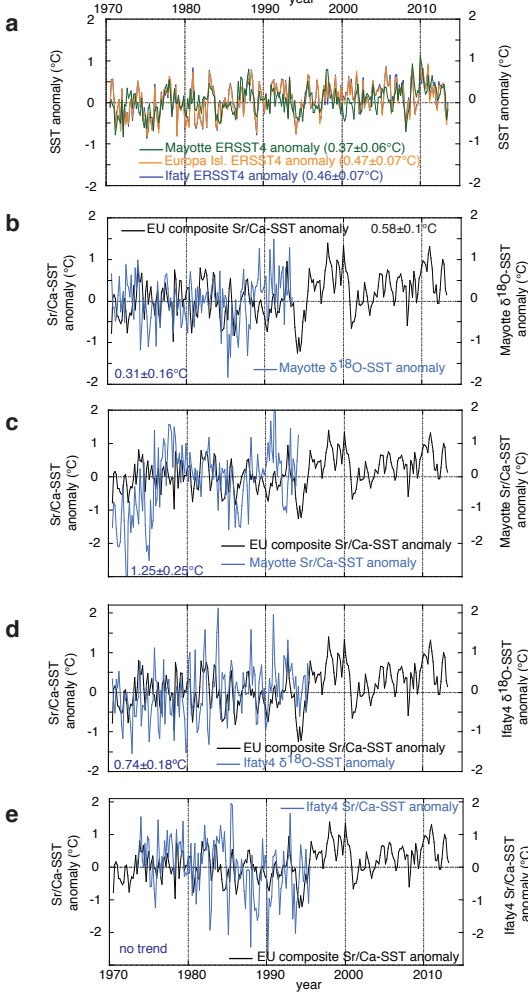

Figure 9 - Regional comparison of Mozambique Channel ERSSTv4 anomalies for Mayotte (green), Europa (orange) and Ifaty Reef, Madagascar (blue) in a) with linear warming trends in brackets. EU Sr/Ca-SST composite anomaly compared with b) Mayotte $\delta^{18}$O-SST anomaly (blue), c) Mayotte Sr/Ca-SST anomaly (blue), d) Ifaty $\delta^{18}$O -SST anomaly (blue) and e) Ifaty Sr/Ca-SST anomaly (blue). Anomalies were calculated for the 1973 to 1993 reference period. Linear warming trends indicated in b) to e) for proxy-SST for individual record length with EU composite Sr/Ca-SST anomaly only indicated once in panel b. Proxy data taken from Zinke et al. (2004, 2008).



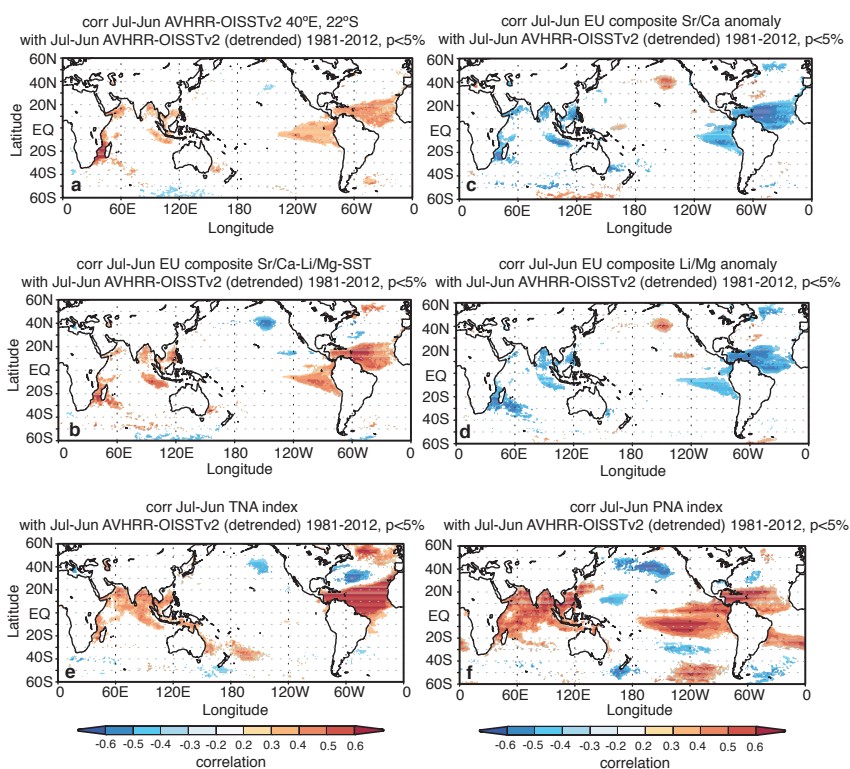

Figure 10 - Spatial correlations of proxy-based coral composite SST reconstructions with local and global AVHRR-OISSTv2 for mean annual data (Banzon et al., 2016). a) local AVHRR-OISSTv2 with global AVHRR-OISSTv2, b) local AVHRR-OISSTv2 with EU composite Sr/Ca-Li/Mg-SST, c) local AVHRR-OISSTv2 with EU composite Sr/Ca ratios and d) local AVHRR-OISSTv2 with EU composite Li/Mg ratios. Panels e and d show spatial correlations of the TNA and PNA indices with global AVHRR-OISSTv2. Only correlations with p<0.05 were coloured.

631