# Peer review of "Multi-trace element sea surface temperature coral reconstruction for the southern Mozambique Channel reveals teleconnections with the tropical Atlantic"

_Biogeosciences, 2018_

## Referee Comment (RC1) · Anonymous Referee #1 · 9 Nov 2018

General comments: Overall, this is an excellent, well-written study by a group of experts in the field of coral-based paleoclimatic reconstructions. The study reports on a new study site on the southeastern coast of Africa, a place with limited instrumental records, thus adding a valuable site to the suite of subannual reconstructions in the Indo-Pacific region. The authors test trace elemental SST proxies, one that is more established (Sr/Ca) and two that are relatively new (Li/Ca and Li/Mg), in two Porites corals of different species. Some studies have criticized coral Sr/Ca as being problematic due to vital effects, yet the direct influence and attribution of these vital effects

remain elusive but some suggest growth-related effects are the source of the problem. Other studies have pointed out that improperly sampling a coral, not along a continuous time growth structure, could be a major reason some studies have problems with the coral Sr/Ca-SST proxy. This study examines these issues in coral-based trace elemental ratio proxies and assesses which is the better SST proxy for their site. They use a careful sampling of the coral skeletal to eliminate any potential suboptimal sampling problems. They also use several different types of temperature data for their proxy assessment including in situ and local air temperature records. They find coral Sr/Ca is the best-performing proxies in their two corals. My only wish is that this study would have also included another newly suggested proxy coral Sr-U for SST proposed by De Carlo et al., 2016, which could be performed simultaneously with their ICP-MS analysis, to assess this other suggested proxy along with the others. The method section (line 190) mentions the precision of U/Ca analysis so it seems U/Ca was measured. Why not add Sr-U to the proxy suite that was assessed?

Specific Comments: I do not have any major issues with this study that would significantly change the overall results. There are a few items that need clarification, suggestions to improve the manuscript, and ways to improve their data analysis.

Move the paragraph in section 3.3 Coral growth parameters and SST to the first section in the Results. I was looking for the extension results to understand why your record is bimonthly with your sampling method. Please give the average annual linear extension for the two corals in this section. In the methods section (lines 167–170) clarify if Coral XDS was performed on a single transect or replicate transects. Were these CoralXDS transects the same as your micro-sampling for geochemical analysis? CoralXDS software should have a reference to the original paper (Helmle et al., 2002, 2011). What were the lowest extension years? Any fall below the Porites threshold of 5 or 6 mm/year? If not, vital effects should not be a problem.

Looking at the X-rays in Supplemental material, I see why you did not sample back further than 2003, shame this coral has such wandering corallite fans. Nice example

of suboptimal corallites. Is this typical of P. mayeri or was this just a small colony?

Methods Line 190-191 The normalization of Sr/Ca with JCP-1, what about Li/Ca and Li/Mg? What is the precision for Li/Mg and Li/Ca?

Figure 4 and Figure S3 Time assignment. There seem to be three clusters of data in all the plots. This is odd, there should not be clusters of data. Double check your Analyseries time assignment, something is off, see next item.

Looking at the residuals plot (Figures 5 and 6) and the methods lines 197–200, I see there is a sinusoidal character in the residuals. This means your error (the residuals) have autocorrelation, which violates the assumptions of Ordinary Least Squares Regression (Durbin and Watson, 1951; Draper and Smith, 1998). Furthermore, the authors assume SST has no error, which in fact it does whether it is bimonthly averaged in situ SST, air temperature, or data products (ERSST and AHVRR SST), thus violating another assumption of OLS regression. I do not see any major issues with your linear regression plots (Figure 4) but if you reverse the regressors, you get a different solution. OLS will either over fit or under fit the proxy data to SST, depending which regressor is the independent regressor. Reduced Major Axis regression is better and takes error in both regressors into account. Weighted least squares regression is even better (York and Evensen, 2004; Thirumalai et al., 2011) since you can vary the error for each data point. However, this switch in regression method should not change your slopes greatly but should reduce your RSME results. The bigger issue is the serial correlation. WLS can take this serial correlation into account in the regression and it increases your confidence intervals, but it would be better to just remove it. The reason the residuals have sinusoidal character is the time assignment was performed with just one point per year (lines 197–200). Another study had this problem was Williams et al. 2012 (see their figures with residuals) where they used one point per year for their time assignment. Williams justified their use of one point per year due to their archive not growing constantly during the year and pausing growth in the winter. Corals should not have this problem in your location. If you use two points per year, this will help remove

the serial correlation from the residuals, 4 points per year are even better. Additionally, make sure you are using the option "select data points" and not default "select interpolate points" in Linage in Analyseries so that not too much interpolation occurs to your data during time assignment (check before and after graphs of your data). This may be why you have data clusters in your scatterplots. This will improve your correlation values and may slightly change your regression equations. Most importantly, the RSME values will change (should be less) and may change proxy performance results. In line 200, mention that the Sr/Ca was used for the time assignment. Was this time assignment applied to the other trace elemental ratios? If so, mention this in your methods section. In your discussion, you should also note using Sr/Ca for the time assignment (whether you use 1, 2, or 4 points per year). Using Sr/Ca for time assignment results in lower RSME values for Sr/Ca compared to the other trace elements. You can convince yourself of this by using Li/Mg or Li/Ca for time assignment and then apply it to Sr/Ca, the Li series should have a lower RSME compared to the other series. This makes it harder to assess which proxy is better since the proxy used for time assignment should have the lowest RSME. You can use another statistic to assess differences between a proxy and temperature, average absolute differences (AD) between the two, which is a robust assessment that does not require the errors to free of autocorrelation or have a normal distribution. The proxy used for time assignment should still have a lower AD but is another test you can perform to assess proxy's performance in reconstruction SST.

Line 192 After doing your own very nice and comprehensive calibration study, why use Correge's –0.0607? Your slopes are lower than his and in my assessment, a better calibration study than Correge's that was just the average for all published studies to date (2006), regardless of the calibration method, SST data used, the possibility of using sub-optimally sampled corals for calibration, etc. Things have vastly improved from 2006 and I would not trust a community average value anymore. See recent paper by (Murty et al., 2018).

Line 464–468, Figure 2 and Methods. It is not mentioned if you cleaned the corals to remove any endolithic algae, tissue layer or biological residue. Corals for Mg analysis are generally cleaned with bleach since Mg is present in biological tissues. In Figure 2, the Mg/Ca is high in the tissue layer, this section should be excluded from the Li/Mg and Mg/Ca data and analysis. Is there any dirty or algae present in the coral skeletons that could be influencing the Mg?

For your regression analysis, why not combined the two corals to see if the results improve?

Table 2 How were degrees of freedom determined? Was if adjusted for serial correlation? For EU3 it seems the Sr/Ca and Li/Mg are repeated in rows 9–12 or is this the average of EU2 and EU3 with the label missing in column 1?

Table 3 What is the second row for EU3 for? Is this EU2 and EU3 averaged together? Table 3 Why are you reporting a standard deviation for RSME? RSME does not have a standard deviation, it is a version of deviation itself between the calculated and measured. RSME is reported as $\pm$ since you take the square root of the sum of the deviations divided by the number of pairs.

Table 4 and all correlations. Give the degrees of freedom (adjusted for serial correlation), # of pairs, and p-values for all correlations.

Figure 2 Can you add the raw trace elemental ratios to the supplemental figures so time assignment can be verified?

Technical Corrections I am not sure the style guide this journal uses, so this may not be an issue, just a British vs. American spelling issue. Dataset is normally two words, data set. Database is one word.

Line 59 Add a reference for trade route statement. I believe the southeastern coast of Africa was part of trade routes before the mid-19th century. ICOADs (V3.1) now goes back to the 1600s (Freeman et al 2017, doi:10.1002/joc.4775) but ERSST stops at

1854. Additionally, there is a newer version of ERSST (V5) that includes new ICOADS data (Huang et. al, 2017 doi: 10.1175/JCLI-D-16-0836.1). Does this version give you any better data coverage for your study site and improve the assessment?

Lines 87–94 There is another coral Li/Mg study (Fowell et al., 2016) that looked at intra-reef variability with a different coral species from the Atlantic and combine Sr/Ca and Li/Mg. That study should be included here but examine the coral sampling in their figure carefully, there were some sub-optimal sampling issues with their corals. The study Montagna et al., 2014 did look at several species of corals, but not the one used by Fowler 2016, so it worth including the Atlantic study in your discussion of previous Li/Mg studies.

Line 154–156 Please give the interval for the air temperature records.

Line 161 Please clarify, is the depth of 12 and 13 m the water depth of the top of the coral colony where the cores were removed, or the water depth of the base of the coral? Did you sample the whole colony? Just trying to see how big these colonies were. Table 1 Add units for Length

References cited: Draper, N.R., Smith, H., 1998. Applied Regression Analysis, Third ed. Wiley-Interscience Publication, New York. Durbin, J., Watson, G.S., 1951. Testing for serial correlation in least squares regression. Biometrika 38, 159-178. Murty, S.A., Bernstein, W.N., Ossolinski, J.E., Davis, R.S., Goodkin, N.F., Hughen, K.A., 2018. Spatial and Temporal Robustness of Sr/Ca-SST Calibrations in Red Sea Corals: Evidence for Influence of Mean Annual Temperature on Calibration Slopes. Paleoceanography and Paleoclimatology 33, 443-456. Thirumalai, K., Singh, A., Ramesh, R., 2011. A MATLAB™ code to perform weighted linear regression with (correlated or uncorrelated) errors in bivariate data. Journ. Geol. Soc India 77, 377-380. York, D., Evensen, N.M., 2004. Unified equations for the slope, intercept and standard errors of the best straight line. Am. J. Phys 72, 367-375.

---

## Referee Comment (RC2) · Anonymous Referee #2 · 22 Nov 2018

Zinke et al. assessed how reliably Sr/Ca, Li/Mg and a combination of both trace element-to-Calcium ratios measured in two (tropical) Indian Ocean Porites sp. colonies serve as SST proxies. In addition, the authors conducted spatial correlation analysis of proxy SST data with instrumental SST.

I have enjoyed reading the ms and – as a non-coral-sclerochronologist – learned a lot. The paper is well-written and organized and certainly merits publication. Data and overall conclusions are fine, but I feel that the presentation and data treatment should be revised (between minor and major revisions).

[Figure]

(1) In my opinion, it is circular to reconstruct SST from trace element-to-Ca data of the same time interval and specimens that were used for proxy calibration. The calibration (+ verification) and reconstruction intervals must be separate, or at least this approach would require different colonies (of both species) for calibration/verification and reconstruction, respectively. One possibility is to limit the calibration interval to 2003-2012 and apply the resulting model then just to the pre-2003 section of core EU3. You could also limit the reconstruction interval to 1970-1980 of EU3, because this time interval was – according to Figure 4 – not used for calibration; both options would still miss a verification interval though. Another possibility is to just present regression models spanning the entire lengths of the cores. Alternatively, the authors use the calibration model by D'Olivio et al. (2018) to estimate SST and compare those data to instrumental SST.

(2) I am missing details on how exactly the Sr/Ca and Li/Mg data were combined. Is the composite Sr/Ca-Li/Mg SST proxy based on a multi-regression model? The equation for this regression model should be provided.

(3) There is some inconsistency regarding the statement on proxy reliability of Sr/Ca and combined Sr/Ca-Li/Mg (compare, e.g., L461-463a and L508/9). Authors should say more clearly (in the main text and in the Abstract) that relative SST changes are reflected equally well by both proxies, but Sr/Ca is still superior to quantify SST. The current Abstract is too vague as the main findings are concerned. It also remained unclear how a combination of Sr/Ca and Li/Mg can "improve SST reconstructions". Table 3 does not support this claim, because RMSE errors are, on average, lower for Sr/Ca-based SST estimates than for those computed from combined Sr/Ca-Li/Mg.

(4) The authors need to better describe the innovative aspect of the study. In particular, they need to highlight (in the Introduction) how their work differs from D'Olivio et al. (2018) and Montagna et al. (2014) (e.g., how much longer were the new coral records in comparison to previous studies; which species were used here and in previous works; how compare the results?).

(5) Linear regression equations may have been the most obvious to use in a mathematical sense, but the actual relationships between trace element-to-Ca ratios (or TE/TE) and SST are nonlinear (compare Gaetani & Cohen 2006; Montagna et al. 2014 etc.). Would it not be better to work with non-linear equations?

(6) I may be worth mentioning somewhere in the Discussion that the Li/Mg serves a more robust thermometer at low temperatures because of the non-linearity of Li/Mg vs. SST relationship.

Other issues: L22: It should read "temperature proxy"

L25-26: Coral = archive, Sr/Ca etc. = proxy. Rephrase sentence accordingly.

L51-53: It should read "The oceans respond". If two full sentences are linked by conjunction, a comma must precede "and".

L54: replace semicolon by comma

L55-58: This sentence needs to be rephrased. Three times "climate" in one sentence. Complicated phrasing.

L76-78: Here you list the slope ranges, but not provide the values revealing the different strengths of the correlations.

L81: "bio-smoothing effects": Explain. Has this expression been used in cited papers?

L201: Tell the reader how long were the calibration intervals in EU3 and EU2.

L226: I would not call 9 years "long-term".

L228/9: "record's lowest ratios" or just "lowest ratios"

L232: I do not understand what you mean: "showed higher values between 2003 and 2012"? Offset relative to what? Why "absolute"?

L233: "Li/Mg between EU2 and EU3...": Replace "between" by "of"

L237: "EU3 showed larger amplitude seasonal variations": Delete "amplitude"

L262/3: "The Sr/Ca and Li/Mg time series of cores EU3 and EU2 were highly consistent in the period of overlap…". What does "consistent" mean here? Rephrase.

L264 (and other occasions in ms): "Li/Mg performed equally well". Odd phrasing; unclear what you mean exactly.

L285: "AVHRR-OISST2 display a more limited seasonality…": Delete one "S" in "OISST"; replace "more limited" by "truncated" or "attenuated".

L291: "reference period" should read "reference periods"

L295: Following Table 2, Sr/Ca-SST slopes are lower than the average reported in Corrège (2006); so this sentence needs to be rephrased.

L321: replace "cooler temperature anomalies" by "smaller anomalies"

Table 2 and elsewhere: Use consistent number of decimals throughout ms (including Figures and Tables), i.e., instead of a slope of "-0.04", write "-0.040".

Figure 2: Comparison would be facilitated if scaling of the y-axes was the same.

Figure 4: ditto for x and y-axes. Also check all other figures.

---

## Author Response (AR1)

**Revision article bg-2018-441 for Biogeosciences**

We like to thank both reviewers for their constructive and very supportive comments. We have uploaded a pdf providing our answers to the reviewers comments. Reviewer questions are indicated in italics and our answer below.

**Reviewer 1:**

*1) My only wish is that this study would have also included another newly suggested proxy coral Sr-U for SST proposed by De Carlo et al., 2016, which could be performed simultaneously with their ICP-MS analysis, to assess this other suggested proxy along with the others. The method section (line 190) mentions the precision of U/Ca analysis so it seems U/Ca was measured. Why not add Sr-U to the proxy suite that was assessed?*

We have tested Sr-U and found that it did not provide reliable SST reconstructions. Furthermore, our aim was to reconstruct bimonthly SST which is currently not possible with Sr-U as it requires averaging multiple data points to estimate temperature for one value and therefore makes it impossible to do direct comparisons of the methods. We have therefore not included such an assessment in the present paper and do not show U/Ca since it does not add to the interpretation. We add some figures here to show the results which we feel do not warrant inclusion, but the coral community might find it helpful since our comments will be published with the paper. Here, our focus is on testing Li/Mg vs. Sr/Ca and their combination for bimonthly time series.

[Figure]

Figure - Left) 3 years averages of EU3 Sr-U (red) compared with NOAA ERSST4 (black). Right) Weighted regression of 3-year averaged Sr-U with ERSST (not significant). Sr-U was calculated following the method in De Carlo et al. (2016). No correlation was found between Sr-U and SST.

*2) Move the paragraph in section 3.3 Coral growth parameters and SST to the first section in the Results.*

Done.

*3) In the methods section (lines 167–170) clarify if Coral XDS was performed on a single*

*transect or replicate transects. Were these CoralXDS transects the same as your micro-sampling for geochemical analysis?*

Transects for CORALXDS were not the same. For geochemistry we have adopted a much more stringent approach to sample the main growth axes. Our age model confirmation with Sr/Ca seasonality was applied to cross-check CORALXDS results, as shown in Supplementary Figures S1 and S2.
We have added the CORALXDS sampling transects to the Supplementary figures S1 and S2 as dashed lines.

*4) What were the lowest extension years? Any fall below the Porites threshold of 5 or 6 mm/year? If not, vital effects should not be a problem.*

None of the years fall below 5mm/yr extension rates, ranging between 7-18mm/yr. Only one year in EU3 (1983) showed 5mm/yr in the CORALXDS data which was corrected with our Sr/Ca extension rate data. We now have included a Supplementary Table S2 listing all annual growth data since 1968.

*5) Methods Line 190-191 The normalization of Sr/Ca with JCP-1, what about Li/Ca and Li/Mg? What is the precision for Li/Mg and Li/Ca?*

All data is referenced to JCp-1, but we also include in coral *in-house* standards for long-term precision of TE/Ca and Li/Mg data. We have adjusted the methods section to align with the former statement and added Sr/Ca, Mg/Ca and Li/Mg values and RSDs in JCp-1 and the in house standard 'Davies Reef'.
It now reads:
"The Sr/Ca, Mg/Ca and Li/Mg data reported here are normalized to the JCp-1 *Porites* sp. standard prepared by the Geological Survey of Japan (Okai et al., 2002) with Sr/Ca = 8.85 mmol/mol ($2\sigma$ RSD = $\pm0.41\%$), Mg/Ca = 4.20 mmol/mol ($2\sigma$ RSD = $\pm0.90\%$) and Li/Mg = 1.47 mmol/mol ($2\sigma$ RSD = $\pm1.04\%$) (N = 17). The Li/Ca data was estimated dividing the Li/Mg by the Mg/Ca data. Long-term reproducibility was determined using the UWA *in-house* Davies Reef coral standard solution with Mg/Ca = $\pm6.24\%$, Sr/Ca = $\pm0.45\%$ and Li/Mg = $\pm1.39\%$ ($2\sigma$ RSD; N = 139) (D'Olivo et al., 2018)."

*6) Figure 4 and Figure S3 Time assignment. There seem to be three clusters of data in all the plots. This is odd, there should not be clusters of data. Double check your Analyseries time assignment, something is off, see next item.*

We checked the linear regressions for monthly vs bimonthly interpolated data with regards to the clusters of data. We found that the clustering increases with decreasing resolution of the SST data, both in terms of spatial (from 0.25° to 2° gridded SST data) and temporal (from monthly to bimonthly) scales. ERSST4 (2°x2° gridded) shows the most extreme clustering, while it is much less for AVHRR OISST. The reason for the clustering is the sudden jumps in

SST between seasons at bimonthly time scales. Our monthly interpolated data don not show clustering (see figure below) confirming our age model. We, however, prefer to work with bimonthly interpolated data since our sampling resolution (6-9 samples per year) is better suited for it.

We have now used 2 anchor points (summer and winter) for our age assignment which did not change the regressions or any conclusions drawn from our work. The methods section has been amended accordingly. Corals never grow constantly between summer and winter in subtropical sites like Europa, the SST seasonality varies between 4.5 and 6°C, so significant SST contrasts between seasons do prevail probably affecting density banding.

[Figure]

*7) Looking at the residuals plot (Figures 5 and 6) and the methods lines 197–200, I see there is a sinusoidal character in the residuals.*

The sinusoidal character of the residuals is most apparent between the 1980-1992 period (Figure 5d). For this particular period, the amplitudes in our TE/Ca SST reconstructions do not match those in instrumental AVHRR-OISST. The latter indicates very low seasonality in several years between 1980 and 1992, far lower than post-1992. It is beyond our capacity to check the quality of AVHRR-OISST in the oldest part of their record. We believe that the mismatch does not relate to the number of anchor points chosen (we have chosen 2 as requested by the reviewer) for calibration. The differences between coral-derived SST and instrumental record reflect either real differences in local SST or a combination of unknown vital effects and temperature.

Figure 6 indicates SST anomalies (seasonality removed) and their residuals (difference between instrumental SST anomaly and coral-derived SST anomaly).

*8) I do not see any major issues with your linear regression plots (Figure 4) but if you reverse the regressors, you get a different solution. OLS will either over fit or under fit the proxy data to SST, depending which regressor is the independent regressor. Reduced Major Axis regression is better and takes error in both regressors into account. Weighted least squares regression is even better (York and Evensen, 2004; Thirumalai et al., 2011) since you can vary the error for each data point. However, this switch in regression method should not change your slopes greatly but should reduce your RSME results. The bigger issue is the serial correlation. WLS can take this serial correlation into account in the regression and it increases your confidence intervals, but it would be better to just remove it.*

Our regression analysis was done in Matlab using the Weighted Least Squares Regression, which we named 'robust regressions' in the figure captions. We apologize for having missed to report this, and have now included it in the methods section. All absolute SST reconstructions are based on WLS regression, including the results for the residuals. Autocorrelation is accounted for in our WLS regressions.

We normally use OLS, and not RMA, because it is the method of choice for coral Sr/Ca-SST regressions (Solow & Huppert, 2004). This is because the relationship of coral Sr/Ca and SST is clearly asymmetric (SST determines coral Sr/Ca, it cannot be the other way round). This is the major criterion for the use of OLS, not the question whether there are errors in the SST observations or not (see e.g. Smith, 2009 "Use and Misuse of the RMA for Line-Fitting" for a discussion). (RMA should be used for symmetric relationships.)

The biggest problems with the application of RMA for coral-Sr/Ca calibrations are the unknown errors. This is nicely summarized in Solow and Huppert (2004). RMA assumes that the error variance in the SST observations equals the error variance of the Sr/Ca determinations. There is no reason to believe that this assumption is warranted. The RMA method can be extended to allow for differences in the error variances (see refs in Solow and Huppert, 2004). To do so, it is necessary to have an estimate of both the SST and Sr/Ca error variance. However, it is practically impossible to determine the error variance of coral Sr/Ca determinations, as these include not only the analytical error but also other factors such as vital effects or skeletal heterogeneities.

However, in our study we opted for weighted least squares regression in line with the suggestion of reviewer 1.

Sr/Ca and Li/Mg were used in age assignments since they are the most robust SST proxies to date. We have not opted for assigning the age model with other TE/Ca ratios. We agree with the reviewer that RMSE will therefore always be better for Sr/Ca and Li/Mg then other TE/Ca ratios.

*9) Line 192 After doing your own very nice and comprehensive calibration study, why use Correge's –0.0607? Your slopes are lower than his and in my assessment, a better calibration study than Correge's that was just the average for all published studies to date (2006), regardless of the calibration method, SST data used, the possibility of using sub-optimally sampled corals for calibration, etc. Things have vastly improved from 2006 and I would not trust a community average value anymore. See recent paper by (Murty et al., 2018).*

We have chosen the Correge slope of -0.061 mmol/mol°C$^{-1}$ for the SST anomaly conversion only because this slope is better suited to look at interannual SST changes at regional space scales (not local) following Gagan et al., 2012. Gagan even suggested to use -0.084 mmol/mol°C$^{-1}$. To our opinion, the -0.061 slope is best suited for SST anomaly conversions as shown in previous work (Pfeiffer et al., 2017). However, all absolute bimonthly SST reconstructions are based on our local regression slopes.

*10) Line 464–468, Figure 2 and Methods. It is not mentioned if you cleaned the corals to remove any endolithic algae, tissue layer or biological residue. Corals for Mg analysis are*

*generally cleaned with bleach since Mg is present in biological tissues. In Figure 2, the Mg/Ca is high in the tissue layer, this section should be excluded from the Li/Mg and Mg/Ca data and analysis. Is there any dirty or algae present in the coral skeletons that could be influencing the Mg?*

We have already reported the cleaning method in the methods section of the original paper in lines 165-166. We have cleaned the corals following the protocol of Nagtegaal et al. (2012). We have excluded the top-most samples from the interpretation since certain TE/Ca ratios still showed an effect despite rigorous cleaning. The latter could well be an effect of the lower densities in the top mm where precipitation efficiency differs from underlying material.

*11) For your regression analysis, why not combined the two corals to see if the results improve?*

Corals EU2 and EU3 have offsets in their mean values, so its not easy to combine them. We have, however, combined proxies when assessing proxy and SST anomalies. Here, we found rather excellent agreement between cores and with SST anomalies.

*12) Table 2 How were degrees of freedom determined? Was if adjusted for serial correlation? For EU3 it seems the Sr/Ca and Li/Mg are repeated in rows 9–12 or is this the average of EU2 and EU3 with the label missing in column 1?*

The regressions applied assume a DOF = N-2, as indicated we used WLS as suggested by the reviewer.

Rows 9 and 11 show the regression parameters for the longer period 1981-2012 for EU3 Sr/Ca and Li/Mg, respectively, with AVHRR-OISSTv2. Rows 10 and 12 show the regression parameters for the period 1970-2012 for EU3 Sr/Ca and Li/Mg, respectively, in relation to ERSST4. This is indicated in the figure caption and periods in last column of Table 2.

*13) Table 3 What is the second row for EU3 for? Is this EU2 and EU3 averaged together? Table 3 Why are you reporting a standard deviation for RSME? RSME does not have a standard deviation, it is a version of deviation itself between the calculated and measured. RSME is reported as since you take the square root of the sum of the deviations divided by the number of pairs.*

The second row for EU3 in Table 3 is for the RMSE for the longer period 1981-2012.
We have omitted the standard deviations, thanks for pointing this out. These numbers indicated the spread in RMSE for individual years over the period considered.

*14) Table 4 and all correlations. Give the degrees of freedom (adjusted for serial correlation), # of pairs, and p-values for all correlations.*

The number of pairs (period in years) and p-values (stars) are indicated in the legend of Table 4. DoF are now added.

*15) Figure 2 Can you add the raw trace elemental ratios to the supplemental figures so time assignment can be verified?*

Done. New Figure S4 added.

*16) Line 59 Add a reference for trade route statement. I believe the southeastern coast of Africa was part of trade routes before the mid-19th century. ICOADs (V3.1) now goes back to the 1600s (Freeman et al 2017, doi:10.1002/joc.4775) but ERSST stops at 1854. Additionally, there is a newer version of ERSST (V5) that includes new ICOADS data (Huang et. al, 2017 doi: 10.1175/JCLI-D-16-0836.1). Does this version give you any better data coverage for your study site and improve the assessment?*

ERSST5 does not provide improvements to ERSST4. In fact, at the coarse resolution of ERSST data (2°x2° gridded) no differences exist between the latest versions of ERSSTv3b, ERSST4 or ERSST5 at the coral reef scale. We have therefore opted for ERSST4 which has been tested more widely in several climatology papers.

*17) Lines 87–94 There is another coral Li/Mg study (Fowell et al., 2016) that looked at intra-reef variability with a different coral species from the Atlantic and combine Sr/Ca and Li/Mg. That study should be included here but examine the coral sampling in their figure carefully, there were some sub-optimal sampling issues with their corals. The study Montagna et al., 2014 did look at several species of corals, but not the one used by Fowler 2016, so it worth including the Atlantic study in your discussion of previous Li/Mg studies.*

Done. We added to discussion:"Applying the -0.097 mmol/mol per °C for Caribbean *Siderastra sidera* forereef corals would underestimate SST anomalies (Fowell et al., 2016)."

*18) Line 154-156 Please give the interval for the air temperature records*

Done.

*19) Line 161 Please clarify, is the depth of 12 and 13 m the water depth of the top of the coral colony where the cores were removed, or the water depth of the base of the coral? Did you sample the whole colony? Just trying to see how big these colonies were. Table 1 Add units for Length*

The depth of 12 and 13 m corresponds to the water depth at the base of the megacolonies. Approximately less than half of the colony EU3 was sampled as we estimate its height to ~3.50 m and its width to 6m. The core top is therefore at 8.5-9.5 m.

**Reviewer 2:**

*1) In my opinion, it is circular to reconstruct SST from trace element-to-Ca data of the same time interval and specimens that were used for proxy calibration. The calibration (+ verification) and reconstruction intervals must be separate, or at least this approach would require different colonies (of both species) for calibration/verification and reconstruction,*

*respectively. One possibility is to limit the calibration interval to 2003-2012 and apply the resulting model then just to the pre-2003 section of core EU3. You could also limit the reconstruction interval to 1970-1980 of EU3, because this time interval was – according to Figure 4 – not used for calibration; both options would still miss a verification interval though. Another possibility is to just present regression models spanning the entire lengths of the cores. Alternatively, the authors use the calibration model by D'Olivio et al. (2018) to estimate SST and compare those data to instrumental SST.*

The calibration periods were all indicated in the figure captions. We have now also included it in methods. We have opted for using the regression model that spans the period of overlap with satellite SST between 1981-2012 for the long core EU3 and 2003-2012 for the short core EU2. Table 2 shows the regressions for different calibration periods. However, our main aim is not the quantification of absolute SST, rather relative changes in SST (anomalies) and how they compare to SST records. We have added a paragraph to the discussion in which we assess the multiproxy method from D'Olivo et al. (2018) applied to our corals.

2) *I am missing details on how exactly the Sr/Ca and Li/Mg data were combined. Is the composite Sr/Ca-Li/Mg SST proxy based on a multi-regression model? The equation for this regression model should be provided.*

We calculated the arithmetic mean between Sr/Ca-SST and Li/Mg-SST.

3) *There is some inconsistency regarding the statement on proxy reliability of Sr/Ca and combined Sr/Ca-Li/Mg (compare, e.g., L461-463a and L508/9). Authors should say more clearly (in the main text and in the Abstract) that relative SST changes are reflected equally well by both proxies, but Sr/Ca is still superior to quantify SST. The current Abstract is too vague as the main findings are concerned. It also remained unclear how a combination of Sr/Ca and Li/Mg can "improve SST reconstructions". Table 3 does not support this claim, because RMSE errors are, on average, lower for Sr/Ca-based SST estimates than for those computed from combined Sr/Ca-Li/Mg.*

We have modified the abstract. It now reads: "In our study, Sr/Ca is still superior to Li/Mg to quantify absolute SST and relative changes in SST."
Regarding the combined Sr/Ca-Li/Mg-SST, Table 3 does show improvement relative to Sr/Ca or Li/Mg only for shorter core record EU2. RMSEs are lower for the combined proxy. The latter hints at the possibility that combined Sr/Ca-Li/Mg can improve SST reconstructions when both proxies work equally well.

We have modified the section in lines 508 onwards and it now reads:
"Overall the results from this study indicated that Sr/Ca is still superior to Li/Mg and was the most reliable SST proxy when applied to a longer time series. However, the excellent agreement between Sr/Ca and Li/Mg and their combination in core EU2 demonstrated that both SST proxies and their combination can provide with greater confidence, more reliable SST reconstructions with lower RMSEs (D'Olivo et al., 2018)."

4) *The authors need to better describe the innovative aspect of the study. In particular, they need to highlight (in the Introduction) how their work differs from D'Olivio et al. (2018) and Montagna et al. (2014) (e.g., how much longer were the new coral records in comparison to previous studies; which species were used here and in previous works; how compare the results?).*

In the Introduction we stated that our record is the first application of the Li/Mg proxy on a multi-decadal scale for *Porites* corals.

5) *Linear regression equations may have been the most obvious to use in a mathematical sense, but the actual relationships between trace element-to-Ca ratios (or TE/TE) and SST are nonlinear (compare Gaetani & Cohen 2006; Montagna et al. 2014 etc.). Would it not be better to work with non-linear equations?*

All regression models for the proxies used in this paper are linear models, which hold for the narrow SST range at our coral site (22-29 °C). This is in line with all tropical corals shown in Montagna et al. 2014 and D'Olivo et al. 2018, which fall on a linear regression line. The exponential relationship for Li/Mg vs SST applies to the entire suite of carbonate secreting archives that grow at vastly different SST.

6) *I may be worth mentioning somewhere in the Discussion that the Li/Mg serves a more robust thermometer at low temperatures because of the non-linearity of Li/Mg vs. SST relationship.*

Done.

*Other issues: L22: It should read "temperature proxy"*
Done.

*L25-26: Coral = archive, Sr/Ca etc. = proxy. Rephrase sentence accordingly.*
Done.

*L51-53: It should read "The oceans respond". If two full sentences are linked by conjunction, a comma must precede "and".*
Done.

*L54: replace semicolon by comma*
Done.

*L55-58: This sentence needs to be rephrased. Three times "climate" in one sentence. Complicated phrasing.*
Word 'climate' removed in one instance

*L76-78: Here you list the slope ranges, but not provide the values revealing the different strengths of the correlations.*
The strength of correlations is not assessed here; it is merely a listing of slopes from the literature.

*L81: "bio-smoothing effects": Explain. Has this expression been used in cited papers?*
This expression has been used by Gagan et al. (2012) and is cited.

*L201: Tell the reader how long were the calibration intervals in EU3 and EU2.*
Calibration periods don't apply here. They are reported in section 3.3 where it is required.

*L226: I would not call 9 years "long-term".*
It refers to the longer period used for anomalies, it reads "Longer time series anomalies were calculated relative to the 1981 to 2010 period."

*L228/9: "record's lowest ratios" or just "lowest ratios"*
Not changed.

*L232: I do not understand what you mean: "showed higher values between 2003 and 2012"? Offset relative to what? Why "absolute"?*
added "relative to EU3.."

*L233: "Li/Mg between EU2 and EU3: : :": Replace "between" by "of"*
Done.

*L237: "EU3 showed larger amplitude seasonal variations": Delete "amplitude"*
Done.

*L262/3: "The Sr/Ca and Li/Mg time series of cores EU3 and EU2 were highly consistent in the period of overlap: : :". What does "consistent" mean here? Rephrase.*
We deleted this sentence.

*L264 (and other occasions in ms): "Li/Mg performed equally well". Odd phrasing; unclear what you mean exactly.*
We deleted this sentence.

*L285: "AVHRR-OISST2 display a more limited seasonality: : :": Delete one "S" in "OISST"; replace "more limited" by "truncated" or "attenuated".*
Done.

*L291: "reference period" should read "reference periods"*
Done.

*L295: Following Table 2, Sr/Ca-SST slopes are lower than the average reported in Corrège (2006); so this sentence needs to be rephrased.*
Latter part of sentence was removed.

*L321: replace "cooler temperature anomalies" by "smaller anomalies"*
We replaced it with 'colder anomalies' because that is what we are trying to convey here.

*Table 2 and elsewhere: Use consistent number of decimals throughout ms (including Figures and Tables), i.e., instead of a slope of "-0.04", write "-0.040".*

We adjusted all slope and p-values to three numbers of decimals.

*Figure 2: Comparison would be facilitated if scaling of the y-axes was the same.*
We deliberately did not scale the y-axes the same because absolute proxy values have an

offset. Plotting all on the same y-axes would make the seasonality harder to spot. We therefore did not re-scale y-axes in Figures 2 and 4. Figure 3 illustrates all proxy versus proxy relationships on the same axes for both cores, complementing Figure 2.

*Figure 4: ditto for x and y-axes. Also check all other figures.*
We deliberately did not scale the y-axes the same because absolute proxy values have an offset. Here, we show the regression plots for both cores reflecting their individual relationship with SST. We therefore did not re-scale this Figure 4.

kind regards,
Zinke and co-authors

**Multi-trace element sea surface temperature coral reconstruction for the southern Mozambique Channel reveals teleconnections with the tropical Atlantic**

Jens Zinke[1,2,3,4], Juan P. D'Olivo[5,6], Christoph J. Gey[2], Malcolm T. McCulloch[5,6], J. Henrich Bruggemann[7], Janice M. Lough[4,5], Mireille M. M. Guillaume[8]

[1]School of Geology, Geography and Environment, University of Leicester, LE17RH, United Kingdom
[2]Institute for Geosciences, Freie Universitaet Berlin, Berlin, 12249, Germany
[3]Molecular and Life Sciences, Curtin University, Perth, WA, Australia
[4]Australian Institute of Marine Science, Townsville, QLD 4810, Australia
[5]The ARC Centre of Excellence for Coral Reefs Studies, Australia
[6]Oceans Graduate School and UWA Oceans Institute, The University of Western Australia, Crawley, WA6009, Australia
[7]UMR ENTROPIE Université de La Réunion-CNRS-IRD, Saint-Denis, France & Laboratoire d'Excellence CORAIL
[8]UMR BOrEA Muséum National d'Histoire Naturelle-SU-UCN-UA-CNRS-IRD, Paris, France & Laboratoire d'Excellence CORAIL

*Correspondence to*: Jens Zinke (jz262@leicester.ac.uk)

**Abstract**

Here we report seasonally resolved sea surface temperatures for the southern Mozambique Channel in the SW Indian Ocean based on multi-trace element temperature proxy records preserved in two *Porites* sp. coral cores. Particularly, we assess the suitability of both separate and combined Sr/Ca and Li/Mg proxies for improved multi-element SST reconstructions. Overall geochemical records from Europa Island *Porites* sp. highlight the potential of Sr/Ca and Li/Mg ratios as high-resolution climate proxies but also show significant differences in their response at this Indian Ocean subtropical reef site. Our reconstruction from 1970 to 2013 using the Sr/Ca-SST proxy reveals a warming trend of 0.58 ± 0.1 °C in close agreement with instrumental data (0.47 ± 0.07 °C) over the last 42 years (1970 to 2013). In contrast the Li/Mg showed unrealistically large warming trends, most probably caused by uncertainties around different uptake mechanisms of trace elements Li and Mg and uncertainties in their temperature calibration. In our study, Sr/Ca is superior to Li/Mg to quantify absolute SST and relative changes in SST. However, spatial correlations between the combined detrended Sr/Ca and Li/Mg proxies compared to instrumental SST at Europa revealed robust correlations with local climate variability in the Mozambique

Channel and teleconnections to regions in the Indian Ocean and southeastern Pacific where surface wind variability appeared to dominate the underlying pattern of SST variability. The strongest correlation was found between our Europa SST reconstruction and instrumental SST records from the northern tropical Atlantic. Only a weak correlation was found with ENSO, with recent warm anomalies in the geochemical proxies coinciding with strong El Niño or La Niña. We identified the Pacific/North American (PNA) atmospheric pattern, which develops in the Pacific in response to ENSO, and the tropical North Atlantic SST as the most likely causes of the observed teleconnections with the Mozambique Channel SST at Europa.

**1 Introduction**

[revised manuscript text omitted]
 (2σ RSD = ±0.41%), Mg/Ca = 4.20 mmol/mol (2σ RSD = ±0.90%) and Li/Mg = 1.47 mmol/mol (2σ RSD = ±1.04%) (N = 17). The Li/Ca data was estimated dividing the Li/Mg by the Mg/Ca data. Long-term reproducibility was determined using the UWA *in-house* Davies Reef coral standard solution with Mg/Ca = ±6.24%, Sr/Ca = ±0.45% and Li/Mg = ±1.39% (2σ RSD; N = 139) (D'Olivo et al., 2018).

The first step after generating the trace element records was to assign an age model. The 2 mm sampling resolution provided 6 to 9 samples per year for any given year and provided robust bimonthly resolved geochemical records. Based on the instrumental SST data from

AVHRR-OISSTv2 and the *in situ* measurements the coldest bimonthly period (on average July/August) and the warmest bimonthly period (on average February/March) for any given year was established for our location. The age model was assigned by simultaneously checking the correlation of Sr/Ca and Li/Mg with the AVHRR-OISSTv2 ensuring the maximum correlation possible was obtained for one proxy without compromising the other proxy. The highest Sr/Ca and Li/Mg annual values in the raw data were tied with the annual minimum and the lowest Sr/Ca and Li/Mg to the annual maximum in the SST records using the open source time series analysis toolkit "Analyseries". Bimonthly (6 samples per year) records were generated by linear interpolation in Analyseries to facilitate comparisons between the different datasets.

[revised manuscript text omitted]

Mg/Ca ratios in core EU3 showed larger seasonal variations than core EU2 (Fig. 3c). EU2 Mg/Ca ratios ranged between 4.24 and 4.61 mmol/mol between 2003 and 2012 while EU3 ranged between 4.40 and 5.36 mmol/mol (4.03 and 5.36 mmol/mol between 1970 and 2012; Fig. 3c, Fig. 4b). EU2 showed lower mean Mg/Ca ratios (~0.4 mmol/mol) than EU3 between 2003 and 2012 (Fig. 3b). Mg/Ca ratios in EU2 and EU3 were significantly correlated, yet lower than Sr/Ca or Li/Mg ($r^2 = 0.34$, p < 0.001, N = 54). Overall, EU3 Mg/Ca showed an increase since 1970 with a marked switch post-2005. EU2 Mg/Ca had no trend.

Li/Ca ratios in core EU2 ranged between 6.03 and 7.20 μmol/mol while in EU3 it ranged between 6.04 and 6.87 μmol/mol (5.97 to 6.87 μmol/mol between 1970 and 2012; Fig. 3d, Fig. 4c). Li/Ca ratios in EU2 and EU3 were significantly correlated ($r^2 = 0.33$, p < 0.001, N = 54), but lower than Sr/Ca or Li/Mg. Li/Ca was positively correlated with Sr/Ca and Li/Mg and negatively with Mg/Ca in both cores for most of the record (Figs. 4c, e, f; Table S1). EU2 Li/Ca largely mirrored variations in Sr/Ca and Li/Mg, while in EU3 Li/Ca showed lower

Jens Zinke 9/1/2019 1:43 PM
Juan Pablo D'Olivo 17/12/2018 4:31 PM

Jens Zinke 9/1/2019 1:55 PM
Jens Zinke 9/1/2019 1:49 PM
Jens Zinke 9/1/2019 1:56 PM

Jens Zinke 9/1/2019 4:42 PM
Jens Zinke 9/1/2019 1:56 PM
Jens Zinke 9/1/2019 1:56 PM
Jens Zinke 13/12/2018 3:51 PM
Jens Zinke 13/12/2018 3:52 PM
Jens Zinke 9/1/2019 1:56 PM
Jens Zinke 9/1/2019 1:56 PM
Jens Zinke 9/1/2019 1:56 PM
Juan Pablo D'Olivo 13/1/2019 4:38 PM
Juan Pablo D'Olivo 13/1/2019 4:39 PM
Juan Pablo D'Olivo 13/1/2019 4:39 PM
Jens Zinke 9/1/2019 1:56 PM
Jens Zinke 9/1/2019 1:56 PM
Juan Pablo D'Olivo 13/1/2019 4:49 PM
Juan Pablo D'Olivo 13/1/2019 4:49 PM
Juan Pablo D'Olivo 13/1/2019 4:49 PM
Jens Zinke 9/1/2019 1:57 PM
Juan Pablo D'Olivo 17/12/2018 4:31 PM

correlations (Table S1). EU3 interannual variability in Li/Ca deviated from the patterns observed in the Sr/Ca, Li/Mg and Mg/Ca data in 1970/71, 1976-78, 1989/90 and between 2001 and 2004 (Fig. 3d). In those years lower EU3 Li/Ca ratios were associated with lower Mg/Ca and higher Sr/Ca and Li/Mg ratios, opposite to the expected relationships (Fig. 3d).

**3.3 Calibration of TE/Ca and SST reconstruction**

Absolute temperature reconstructions were obtained from the regression of the bimonthly Sr/Ca and Li/Mg ratios with the AVHRR-OISSTv2 and ERSSTv4 data (Fig. 5; Table 2; for ERSSTv4 see Fig. S5 and for Mg/Ca and Li/Ca vs. SST see Fig. S6). Both of the coral datasets showed highly significant ($p < 0.001$) correlation coefficients with the temperature products over the period of overlap (2003 to 2012) with $r^2_{EU3\ Sr/Ca} = 0.92$, $r^2_{EU2\ Sr/Ca} = 0.93$, $r^2_{EU3\ Li/Mg} = 0.78$ and $r^2_{EU2\ Li/Mg} = 0.93$ (Table 2). Correlation coefficients of EU3 Sr/Ca and Li/Mg for the longer periods 1981 to 2012 and 1970 to 2012 with AVHRR-OISSTv2 and ERSSTv4, respectively, were also high (Table 2; Fig. S5). The regression slope of TE ratios with the two SST products varied between -0.040 and -0.051 mmol/mol per °C for Sr/Ca and between -0.045 and -0.064 mmol/mol per °C for Li/Mg (Table 2). Overall, the regression slopes were marginally lower for regressions with AVHRR-OISSTv2 compared to ERSSTv4. Weighted least squares regressions with 1.5 years *in situ* SST data between 2009 and 2010 revealed similar regression slopes for Sr/Ca and Li/Mg but with narrower range (-0.042 to -0.047 mmol/mol per °C for Sr/Ca and –0.045 to -0.052 mmol/mol per °C for Li/Mg) and lower correlation coefficients ($r^2_{EU3\ Sr/Ca} = 0.70$, $r^2_{EU2\ Sr/Ca} = 0.76$, $r^2_{EU3\ Li/Mg} = 0.73$ and $r^2_{EU2\ Li/Mg} = 0.81$). All correlations were statistically significant with $p < 0.050$.

The maximal seasonal range over the period 1970 to 2012 of the reconstructed bimonthly Sr/Ca-SST and Li/Mg-SST varied between 22 °C and 30 °C in both cores with a mean seasonal amplitude of $4.33 \pm 0.67$ °C (Fig. 5; for ERSSTv4 see Fig. S7) in close agreement with *in situ* SST ($4.82 \pm 0.05$ °C for 2009 to 2010) and regional AVHRR-OISSTv2 ($4.67 \pm 0.7$ °C for 1981 to 2013) and ERSSTv4 ($4.52 \pm 0.44$ °C for 1970 to 2012).

Residuals (calculated as the difference between coral-derived SST and AVHRR-OISSTv2 for individual record length) are presented in Fig. 6 and RMSE's in Table 3 (for ERSSTv4 see Fig. S7). The coral Sr/Ca and Li/Mg-SST reconstructions had the lowest residuals between 1993 and 2012 with AVHRR-OISSTv2, with slightly larger residuals prior to 1993 (core

Jens Zinke 9/1/2019 1:57 PM

Jens Zinke 9/1/2019 1:57 PM

Jens Zinke 3/12/2018 10:37 AM

Jens Zinke 9/1/2019 1:57 PM

Jens Zinke 9/1/2019 1:58 PM

Jens Zinke 9/1/2019 1:58 PM

Jens Zinke 13/12/2018 3:56 PM

Jens Zinke 9/1/2019 1:58 PM

Jens Zinke 19/12/2018 10:58 AM

Jens Zinke 9/1/2019 4:45 PM

Jens Zinke 9/1/2019 4:45 PM

Jens Zinke 9/1/2019 2:00 PM

Jens Zinke 19/12/2018 11:52 AM

Jens Zinke 9/1/2019 2:00 PM

Jens Zinke 9/1/2019 2:01 PM

[revised manuscript text omitted]

Juan Pablo D'Olivo 13/1/2019 4:59 PM

Jens Zinke 9/1/2019 2:08 PM

Jens Zinke 9/1/2019 2:08 PM

Juan Pablo D'Olivo 13/1/2019 5:00 PM

Jens Zinke 15/1/2019 3:23 PM

Jens Zinke 15/1/2019 3:23 PM

Juan Pablo D'Olivo 13/1/2019 5:00 PM

Juan Pablo D'Olivo 13/1/2019 5:01 PM

The regressions of detrended seasonal averages in AVHRR-OISSTv2 for Europa with the Niño3.4 index of ENSO variability showed weak, yet statistically significant correlations in the season from February to April (r = 0.47; p < 0.010; Table 4). The correlations between ERSSTv4 and Niño 3.4 were weaker (r = 0.34, p < 0.050; Table 4). The detrended coral composite Sr/Ca, Li/Mg and their combined SST reconstructions showed no significant correlations with the Niño3.4 index. However, the Pacific/North American (PNA) pattern (Wallace & Gutzler, 1981), which is an atmospheric response to ENSO, showed statistically significant correlations with AVHRR-OISSTv2 (r =0.67, p < 0.001) for Europa and the coral-derived SST anomalies between 1981 and 2012 (r = 0.42, p = 0.014; Table 4). The spatial correlation pattern of the PNA index with global AVHRR-OISSTv2 revealed a similar pattern as observed for the coral-based SST (Fig. 10).

**4 Discussion**

**4.1 Reliability of Sr/Ca and Li/Mg as SST proxies**

[revised manuscript text omitted]

strategies. For example, D'Olivo *et al.*, (2018) showed a deviation for *Porites solida* from other massive *Porites* species in the relationships between Sr/Ca and Li/Mg with temperature. However, this requires further investigations as this is the first long-term (multi-decadal) reconstruction based on massive *Porites*. Despite these uncertainties Li/Mg was overall the second-best performing proxy in this study with the detrended Li/Mg data showing an excellent agreement with the instrumental SST and Sr/Ca data. Furthermore, the interannual and decadal SST variations as well as spatial correlation patterns in the Li/Mg appeared not to have been affected and can be interpreted with high confidence as indicated by our field correlations. Overall the results from this study indicated that Sr/Ca is still superior to Li/Mg and was the most reliable SST proxy when applied to a longer time series. However, the excellent agreement between Sr/Ca and Li/Mg and their combination in core EU2 demonstrated, that both SST proxies and their combination can provide with greater confidence, more reliable SST reconstructions with lower RMSEs (D'Olivo et al., 2018).

In addition, we tested the multiproxy paleothermometer based on Sr/Ca and Li/Mg data proposed by D'Olivo et al., (2018) in both corals. Using the proposed equations on cores EU2 and EU3 their respective RMSE values were 1.1 °C and 0.2 °C higher than the values obtained from a direct calibration against the SST products. While the multiproxy method produced excellent results for core EU3, the higher RMSE for EU2 could be explained by the limitations of the multiproxy method as solutions appear to deteriorate *for Porites solida* and corals with small effective tissue thickness (~ < 4 months). In this case EU2 had an effective tissue thickness of 3.9 months and was identified as *P. solida* while EU3 had an effective tissue thickness of 9.5 months and was identified as P. *mayeri*. This confirms the effectiveness of the multiproxy method proposed by D'Olivo et al., (2018) when applied within its limits.

[revised manuscript text omitted]

De Villiers, S., Nelson, B. K., and Chivas A. R.: Biological controls on coral Sr/Ca and delta$^{18}$O reconstructions of sea surface temperatures, Science, 269, 1247-1249, 1995.

Enfield, D. B., Mestas, A. M., Mayer, D. A., and Cid-Serrano. L: How ubiquitous is the

dipole relationship in tropical Atlantic sea surface temperatures?, J. Geophys. Res., 104, 7841-7848, 1999.

Freeman, E., Woodruff, S. D., Worley, S. J., Lubker, S. J., Kent, E. C., Angel, W. E., Berry, D. I., Brohan, P., Eastman, R., Gates, L., Gloeden, W., Ji, Z., Lawrimore, J., Rayner, N. A., Rosenhagen, G., and Smith, S. R.: ICOADS Release 3.0: a major update to the historical marine climate record, Int. J. Climatol., 37, 2211-2232, 2017.

Fowell, S. E., Sandford, K., Stewart, J. A., Castillo, K. D., Ries, J. B., and Foster, G. L.: Intrareef variations in Li/Mg and Sr/Ca sea surface temperature proxies in the Caribbean reef-building coral *Siderastrea siderea*, Paleoceanography, 31, 1315–1329, 2016.

[revised manuscript text omitted]

Van den Berg, M. A., Morris, T., and Roberts, M. J.: Long-term temperature monitoring in the Mozambique Channel. 5th Western Indian Ocean Marine Science Association scientific symposium, Durban, South Africa, 22-26 October 2007, abstract, 2007.

Varela, R., Álvarez, I., Santos, F., deCastro, M., and Gómez-Gesteira, M.: Has upwelling strengthened along worldwide coasts over 1982-2010?, Scientific Reports, 5, 10016, doi:10.1038/srep10016, 2015.

Woodruff, S. D., Worley, S. J., Lubker, S. J., Ji, Z., J. Freeman, E., Berry, D. I., Brohan, P., Kent, E. C., Reynolds, R. W., Smith, S. R., and Wilkinson, C.: ICOADS Release 2.5: Extensions and enhancements to the surface marine meteorological archive, Int. J. Climatol., 31, 951-967, 2011.

Xie, S. P., Deser, C., Vecchi, G., Ma, J., Teng, H., and Wittenberg, A. T.: Global warming pattern formation: Sea surface temperature and rainfall, J. Climate, 23, 966-986, 2010.

Zinke, J., Reuning, L., Pfeiffer, M., Wassenburg, J.A., Hardman, E., Jhangeer-Khan, R., Davies, G.R., Ng, C.K.C., and Kroon, D.: A sea surface temperature reconstruction for the southern Indian Ocean trade wind belt from corals in Rodrigues Island (19S, 63E), Biogeosciences, 13, 5827-5847, 2016.

Zinke, J., Hoell, A., Lough, J., Feng, M., Kuret, A., Clarke, H., Ricca, V., and McCulloch, M. T.: Coral record of southeast Indian Ocean marine heatwaves with intensified Western Pacific temperature gradient, Nat. Comm., 6, 8562, doi: 10.1038/ncomms9562, 2015.

Zinke, J., Timm, O., Pfeiffer, M., Dullo, W.-C., Kroon, D., and Thomassin, B. A.: Mayotte coral reveals hydrological changes in the western Indian between 1865 to 1994, Geophys. Res. Lett., 35, L23707, doi:10.1029/2008GL035634, 2008.

Zinke, J., Dullo, W.-C., Heiss, G. A., and Eisenhauer, A.: ENSO and subtropical dipole variability is recorded in a coral record off southwest Madagascar for the period 1659 to 1995, Earth Planet. Sci. Lett., 228 (1-2), 177-197, 2004.

**Tables**

| Location | Latitude (S) | Longitude (E) | Depth (m) | Core # | Length | Mean Extension mmyr$^{-1}$ | Collection dates |
|---|---|---|---|---|---|---|---|
| **North Reef** | 22°19.839 | 40°21.758 | 12.80 | EU-2 | 105 | 1.07±0.19 | 2/5/2013 |
| **North-East Reef** | 22°20.119 | 40°23.333 | 12.00 | EU-3 | 136 | 1.20±0.27 | 3/5/2013 |

Table 1 – Coral core GPS locations from Europa, water depth, core name, core length, mean extension rate between 1968 and 2012 (standard deviation in brackets) and collection dates.

[revised manuscript text omitted]